# ON FINE-GRAINED I/O COMPLEXITY OF ATTENTION BACKWARD PASSES

## ABSTRACT

Large Language Models (LLMs) have demonstrated remarkable capabilities in processing long-context information. However, the quadratic complexity of attention computation with respect to sequence length poses significant computational challenges, and I/O aware algorithms have been proposed. This paper presents a comprehensive analysis of the I/O complexity for attention mechanisms, focusing on backward passes by categorizing them into small and large cache scenarios. Using the red-blue pebble game framework, we establish tight bounds on I/O complexity across all cache sizes. We confirm that the de facto standard I/O aware algorithm FlashAttention is optimal for both forward and backward passes for the large cache size scenario. For small cache sizes, we provide an algorithm that improves over existing methods and achieves tight bounds. Additionally, we extend our analysis to sparse attention, a mainstream speeding-up approach, deriving fine-grained lower bounds for both forward and backward passes and both small and large caches. Our findings complete the theoretical foundation for I/O complexity in attention mechanisms, offering insights for designing efficient algorithms of LLM training and inference.

## 1 INTRODUCTION

Large Language Models (LLMs), such as GPT-4 (Achiam et al., 2023), Claude (Anthropic, 2024), Llama (Llama Team, 2024), and more recently o1 (OpenAI, 2024) from OpenAI, have demonstrated immense potential to enhance various aspects of our daily lives, including conversational AI (Liu et al., 2024), AI agents (Xi et al., 2023; Chen et al., 2024b), search AI (OpenAI, 2024), AI assistants (Kuo et al., 2024; Feng et al., 2024b), and many others. One of the most emergent abilities of LLMs is dealing with long-context information, which is crucial for processing materials such as academic papers, official reports, and legal documents. LLMs have proven adept at tackling long-context tasks, such as zero-shot summarization (Chhabra et al., 2024; Zhao et al., 2024) and maintaining very long-term conversations (Xu et al., 2022; Maharana et al., 2024). OpenAI's o1 model (OpenAI, 2024) serves as a significant advancement in this area. It leverages Chain-of-Thought (CoT) reasoning (Wei et al., 2022; Kojima et al., 2022) and employs Retrieval Augmented Generation (RAG) (Lewis et al., 2020; Gao et al., 2023) to exhibit PhD-level abilities, where both techniques require long context inputs for generation. This proficiency underscores the necessity for developing long-context modeling capabilities within LLMs.

LLMs are primarily based on the Transformer architecture (Vaswani et al., 2017), whose core component is the self-attention mechanism. However, the quadratic complexity of attention computation with respect to sequence length dominates the computational FLOPs during long-context training and inference. To address this issue, FlashAttention (Dao et al., 2022; Dao, 2023; Shah et al., 2024) accelerates attention computation and has become the de facto standard in the industry of LLM training and inference deployment. The success of FlashAttention lies in its I/O awareness (Aggarwal & Vitter, 1988), accounting for reads and writes to different levels of fast *cache* (e.g., GPU on-chip SRAM) and slow *memory* (e.g., GPU high-bandwidth memory) within the hardware hierarchy. Leveraging modern hardware design in GPUs, e.g., NVIDIA A100 and H100, efficiently allows FlashAttention to be integrated as a go-to method for LLM training and inference.

For the I/O complexity of exact attention[1] forward computation, the theoretical analysis of FlashAttention in Dao et al. (2022) only provides upper and lower bounds when the cache size $M \in [d, nd]$. Their bounds are only tight in the range of $M = \Theta(nd)$, where $n$ is the input sequence length and $d$ is the hidden dimension. By fine-grained analysis, a recent work (Saha & Ye, 2024) provides matching upper and lower I/O complexity bounds of the attention *forward* passes for *any* cache size $M$. For the I/O complexity of attention *backward* passes, existing work only provides an upper bound for FlashAttention for the cache size $M \in [d, nd]$ (Dao et al., 2022), without known lower bounds. Thus, the tight bounds for the I/O complexity of attention backward passes are lacking. This raises a natural question:

*What is the optimal I/O complexity of attention backward computations for any cache size?*

In this paper, we address this question and provide matching upper and lower I/O complexity bounds for backward passes of exact attention computation for all cache sizes, completing the picture of I/O complexity for the attention mechanism.

## 1.1 OUR CONTRIBUTIONS

In this work, we analyze the I/O complexity in the same setting as the existing work of FlashAttention (Dao et al., 2022) and Saha & Ye (2024). We consider a two-level memory hierarchy consisting of a small but fast layer called the *cache* and a large but slower layer referred to as *memory*. The I/O complexity quantifies the data transfer between these two layers, which can be formally defined as a red-blue pebble game (Hong & Kung, 1981) as in Definition 3.4. We study the exact attention computation using standard matrix multiplication as the existing work[2] and focus on backward gradient computation. We establish matching I/O complexity upper and lower bounds for attention backward computation (formalized in Theorem 1.1 and illustrated in Fig. 1). Combined with the attention forward results from Saha & Ye (2024), this completes the theory of I/O complexity in the attention mechanism. Our main result is stated as follows:

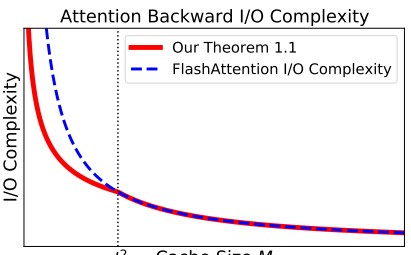

Figure 1: Attention backward I/O complexity comparison. The $x$-axis is the cache size, and the $y$-axis is the I/O complexity. The red line represents our tight upper/lower bound (Theorem 1.1), and the blue dash denotes the upper bound for FlashAttention (Dao et al., 2022). The cross point is $M = \Theta(d^2)$, the dividing point of large cache and small cache settings. The results show that FlashAttention is optimal when $M = \Omega(d^2)$.

**Theorem 1.1** (Main result). *Let $n$ be the sequence length, $d$ the head dimension, and $M$ the cache size. The I/O complexity of attention backward computation under standard matrix multiplication is* $\Theta\left(\min\left\{\frac{n^2 d^2 + n d^3}{M}, \frac{n^2 d + n d^2}{\sqrt{M}}\right\}\right)$.

To interpret our main result, we categorize the cache size $M$ into two cases: the small cache case where $M = o(d^2)$ and the large cache case where $M = \Omega(d^2)$ (see Fig. 1 for illustration).

In the small cache scenario, $M = o(d^2)$, by computation graph Fig. 2 and Algorithm 6, we show that the upper bound of the I/O complexity is $O(\frac{n^2 d + n d^2}{\sqrt{M}})$. In detail, Algorithm 6 explicitly read/write the $n \times n$ attention matrix and other $n \times d$ intermediate matrices from/to memory. Note that, when $M = o(d^2)$, our Algorithm 6 has a better upper bound than FlashAttention, whose upper bound is $O(\frac{n^2 d^2 + n d^3}{M})$. Furthermore, to establish a lower bound on the I/O complexity, we show that the I/O complexity of attention backward computation is equivalent to the I/O complexity of matrix multiplication when $M = o(d^2)$, which matches the upper bound of Algorithm 6.

In the more practical large cache case, $M = \Omega(d^2)$, we prove an upper bound $O(\frac{n^2 d^2 + n d^3}{M})$ on the I/O complexity for the attention backward algorithms (Algorithm 9), which matches that of

---

[1]In this work, we only consider exact attention computation without any approximation.

[2]Note that there are many fast matrix multiplication methods. We do not study them, as they are hard to be parallelized. Standard matrix multiplication is still the most popular implementation on GPU, e.g., PyTorch. We refer readers to Section 3 for more details.

Table 1: Summary of our contributions. We categorize the cache size $M$ into two cases: (1) Large cache $M = \Omega(d^2)$; (2) Small cache $M = o(d^2)$. Assume $n \geq d$. We list our contributions for general and sparse attention below. $Z_{\text{input}}$ and $Z_{\text{QK}}$ denote the number of nonzero entries of the input matrix and the key-query matrix, respectively.

| Attention Algorithm | | Large Cache | Reference | Small Cache | Reference |
|---|---|---|---|---|---|
| General | Forward Upper | $O(n^2 d^2/M)$ | Dao et al. (2022) | $O(n^2 d/\sqrt{M})$ | Saha & Ye (2024) |
| | Forward Lower | $\Omega(n^2 d^2/M)$ | Saha & Ye (2024) | $\Omega(n^2 d/\sqrt{M})$ | Saha & Ye (2024) |
| | Backward Upper | $O(n^2 d^2/M)$ | Dao et al. (2022) | $O(n^2 d/\sqrt{M})$ | Theorem 4.3 |
| | Backward Lower | $\Omega(n^2 d^2/M)$ | Theorem 4.2 | $\Omega(n^2 d/\sqrt{M})$ | Theorem 4.4 |
| Sparse | Forward Lower | $\Omega(Z_{\text{input}}^2/M)$ | Theorem 4.5 | $\Omega(Z_{\text{input}}\sqrt{Z_{\text{QK}}}/\sqrt{M})$ | Theorem 4.5 |
| | Backward Lower | $\Omega(Z_{\text{input}}^2/M)$ | Theorem 4.5 | $\Omega(Z_{\text{input}}\sqrt{Z_{\text{QK}}}/\sqrt{M})$ | Theorem 4.5 |

FlashAttention (Dao et al., 2022; Dao, 2023; Shah et al., 2024). We prove that this upper bound is tight by providing a matching lower bound for the I/O complexity of attention backward using the red-blue pebble game analysis framework from Hong & Kung (1981).

Therefore, we provide the optimal bounds and algorithms for backward passes for all cache sizes. This fully characterizes the I/O complexity of attention forward/backward when combined with existing results on forward passes (Saha & Ye, 2024). Notably, we confirm that FlashAttention is optimal for both the forward and backward passes when the cache size is large enough $M = \Omega(d^2)$. Moreover, in recent years, sparse attention has become another mainstream method for speeding up the training process of transformer-based models (Child et al., 2019; Zaheer et al., 2020; Beltagy et al., 2020). These approaches mainly focus on techniques for sparsifying the attention matrix, thereby reducing the quadratic bottleneck in running time. However, it remains unknown whether this method can be integrated with I/O-aware algorithms like FlashAttention. Consequently, we further analyze the I/O complexity of sparse attention to provide theoretical guarantees, offering fine-grained lower bounds.

**Theorem 1.2** (Lower bound for sparse attention forward and backward, informal version of Theorem 4.5). *Let $Z_{\text{input}}$ and $Z_{\text{QK}}$ be the number of nonzero entries of the input matrix and the key-query matrix, respectively. Then, any algorithm for both attention forward and backward computation using sparse semi-ring matrix multiplication has I/O complexity*

$$\Omega\left(\min\left\{\frac{Z_{\text{input}}^2}{M}, \frac{Z_{\text{input}}\sqrt{Z_{\text{QK}}}}{\sqrt{M}}\right\}\right).$$

Our I/O complexity lower bound for sparse attention recovers the lower bound for both attention forward and backward passes when matrices involved in attention computation are dense, i.e., $Z_{\text{input}} = \Omega(nd), Z_{\text{QK}} = \Omega(n^2)$. In such case, our lower bound reads as $\Omega(\min\{\frac{n^2 d^2}{M}, \frac{n^2 d}{\sqrt{M}}\})$, matching Theorem 1.1. The dividing point between small and large cache for sparse attention is $M = Z_{\text{input}}^2/Z_{\text{QK}}$, which also matches the dense case.

We summarize our contributions in Table 1 and also conclude as follows:

- For small cache sizes $M = o(d^2)$ in the backward pass, we present optimal upper and lower bounds and propose an algorithm achieving the optimal (Algorithm 6). Notably, FlashAttention is not optimal in this setting, and our algorithm outperforms it.

- For large cache sizes $M = \Omega(d^2)$ in the backward pass, we establish an optimal lower bound that matches the existing upper bound. We also prove the optimal upper bound and introduce an optimal algorithm (Algorithm 9), matching the existing results for FlashAttention but providing a different analysis.

- For sparse attention, we offer fine-grained lower bounds for both forward and backward passes and across all cache sizes (Theorem 4.5).

## 2    RELATED WORK

**Learning with Bounded Memory and I/O Complexity.** A common memory model in computational systems is the two-level memory hierarchy. In this model, there are two layers of memory: a small but fast layer called the *cache*, and a large but slower layer called the *memory*. The I/O (input/output) complexity of an algorithm measures its efficiency based on the number of data transfer operations it performs between the cache and the memory. The early work of Hong & Kung (1981) formulated the I/O complexity mathematically using the language of graph theory. Learning with bounded memory has been studied in various fields in machine learning such as online learning (Srinivas et al., 2022; Peng & Rubinstein, 2023; Peng & Zhang, 2023), convex optimization (Marsden et al., 2022; Chen & Peng, 2023), active learning (Hopkins et al., 2021), attention computation (Addanki et al., 2023), and continual learning (Chen et al., 2022; Ermis et al., 2022).

Due to space constraints, we move some related works to the Appendix A.

## 3    PRELIMINARY

In this work, we consider using a standard algorithm for matrix multiplication, which means that for any two matrices $A \in \mathbb{R}^{n_1 \times d}, B \in \mathbb{R}^{d \times n_2}$, each entry of $AB$ is computed by $(AB)_{i,j} = \sum_{k=1}^{d} A_{i,k} B_{k,j}$ for $i \in [n_1], j \in [n_2]$. Note that this setting is also used in FlashAttetnion (Dao et al., 2022) and Saha & Ye (2024). Although certain "fast" algorithms, such as Strassen (Strassen, 1969) or Coppersmith–Winograd (Coppersmith & Winograd, 1987), have lower asymptotic computational complexity, they typically incur significantly large hidden constants and do not parallelize as efficiently on modern GPUs. In practice, libraries such as cuBLAS and CUTLASS optimized for standard GEMMs (General matrix multiplications), often outperform any known fast-matrix approach on the matrix sizes relevant to deep learning. Therefore, assuming standard matrix multiplication offers an accurate reflection of how attention computation is commonly carried out in real-world systems. Now, we introduce some key concepts needed for this paper.

### 3.1    KEY CONCEPT OF ATTENTION

Before formally stating our results, we begin by precisely defining the problems we study. We define the following computation of the general Softmax attention forward layer.

**Definition 3.1** (Attention forward computation). *Let $n$ be the input length and $d$ be the head dimension. Let $A_1, A_2, A_3 \in \mathbb{R}^{n \times d}$ be the inputs of previous layer. Given query, key and value weights matrix $W_Q, W_K, W_V \in \mathbb{R}^{d \times d}$, we have the Softmax attention forward computation being*

$$\mathsf{Attn}(A_1, A_2, A_3) := D^{-1} \exp(A_1 W_Q W_K^\top A_2^\top) A_3 W_V,$$

*where (1) $D := \mathrm{diag}(\exp(A_1 W_Q W_K^\top A_2^\top) \cdot \mathbf{1}_n)$, (2) $\exp$ denotes the exponential function and is applied entry-wisely, (3) $\mathrm{diag}()$ operation takes a vector and outputs a diagonal matrix with the entries of that vector, and (4) $\mathbf{1}_n$ denotes the length-$n$ all ones vector.*

*To simplify and focus more clearly on the core computational aspects of the problem, we set $X = W_Q W_K^\top \in \mathbb{R}^{d \times d}$ and $Y = W_V \in \mathbb{R}^{d \times d}$.*

Note that, we have

$$\mathsf{Softmax}(A_1 X A_2^\top) = D^{-1} \exp(A_1 X A_2^\top) \in \mathbb{R}^{n \times n},$$

and usually we call it the attention matrix. The above definition is general and encompasses both self-attention and cross-attention mechanisms in Transformer architectures. Specifically, self-attention occurs when $A_1 = A_2 = A_3$, meaning that the queries, keys, and values are all derived from the same source. In contrast, cross-attention happens when $A_2 = A_3$, indicating that the keys and values come from one source while the queries come from the other.

Notably, FlashAttention (Dao et al., 2022; Dao, 2023; Shah et al., 2024) and Saha & Ye (2024) consider $Q, K, V \in \mathbb{R}^{n \times d}$ after applying the linear layer to the previous inputs, while we consider a more detailed structure as $Q = A_1 W_Q, K = A_2 W_K, V = A_3 W_V$ (Definition 3.1) explicitly calculating module-wise gradients on attention weights. This explains why our I/O complexity

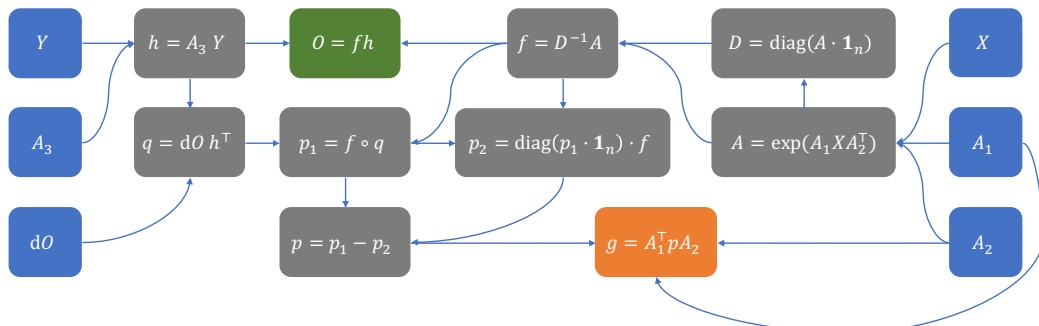

Figure 2: The computational graph for attention forward and backward. The blue boxes are input matrices, the gray boxes are intermediate matrices, the green box is the forward output, and the orange box is the final gradient matrix. Here, $A_1, A_2, A_3$ denote the previous inputs, $\mathrm{d}O$ denotes the upstream gradient, and $X, Y$ denote the attention weights. More detailed definitions of each variables can be found in Section 3 and B.

bound $\Theta(\min\{\frac{n^2d^2+nd^3}{M}, \frac{n^2d+nd^2}{\sqrt{M}}\})$ in Theorem 1.1 has an additional term $nd^2$ in the small cache case and $nd^3$ in the large cache case. When $n \geq d$, the additional term will disappear.

Mathematically, optimizing the attention computation involves adjusting the attention weight matrices $X$, and $Y$. Using the previous results on attention gradients from Alman & Song (2024a) and Liang et al. (2024c), we have the following definition of attention gradient:

**Definition 3.2** (Attention backward gradient). *Let $A_1, A_2 \in \mathbb{R}^{n \times d}$. Let $p(X) \in \mathbb{R}^{n \times n}$ be defined in Definition B.9 (see Fig. 2 for an illustration). Let $L(X)$ be some loss function. The attention backward gradient for $X \in \mathbb{R}^{d \times d}$ is $\frac{\mathrm{d}L(X)}{\mathrm{d}X} = A_1^\top p(X) A_2$.*

**Remark 3.3.** *Since the attention module depends only linearly on $Y$, it is straightforward to incorporate it into an algorithm, and it is not a complexity bottleneck. Thus, we focus on the case where $X$ is variable and $Y$ is a fixed input.*

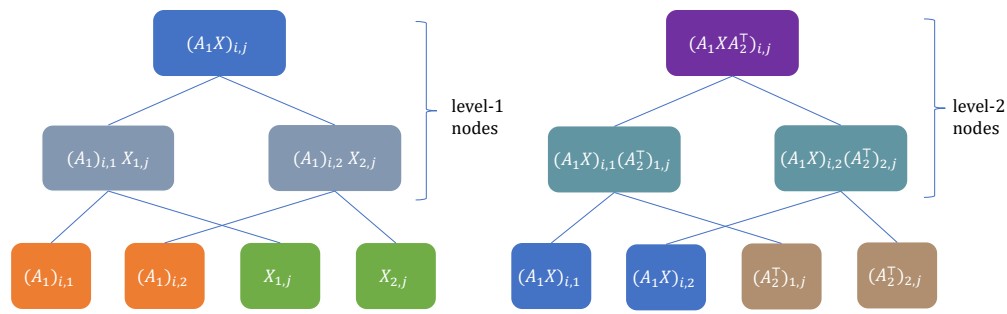

Figure 3: This diagram shows a summation tree with $d = 2$ in the computational graph for the backward passes of attention using standard matrix multiplication. The orange and green nodes represent the input nodes of the level-1 summation tree. The brown nodes, along with the blue nodes (output from the level-1 summation tree), serve as inputs for the level-2 summation tree. The purple nodes represent the target output. When $d$ gets larger, the summation tree will expand with additional layers, where each new layer introduces intermediate nodes that represent the sums of pairs of nodes from the previous layer, i.e., there will be a total $1 + \log_2 d$ layer in total.

## 3.2 SUMMATION TREE

In this subsection, we need to introduce the computational graph of the attention backward gradient, which is the key concept in our I/O complexity analysis.

In the computational graph shown in Fig. 2, we can first compute $A_1 X$ and then compute $(A_1 X) A_2^\top$, or first compute $X A_2^\top$ and then compute $A_1 (X A_2^\top)$. In either case, we perform two matrix multi-

plications: one between an $n \times d$ matrix and a $d \times d$ matrix, and the other between an $n \times d$ matrix and a $d \times n$ matrix. Without loss of generality for illustration, we consider the first case. To compute $A_1 X$, we need to calculate the products $\{(A_1)_{i,k} X_{k,j}\}$ for all $i \in [n]$, $k \in [d]$, $j \in [d]$. Each entry $(A_1 X)_{i,j}$ is then obtained by summing these products over $k$: $(A_1 X)_{i,j} = \sum_{k=1}^{d} (A_1)_{i,k} X_{k,j}$. In the computational graph, this summation is represented by a summation tree that connects the product nodes $(A_1)_{i,k} X_{k,j}$ to the sum node $(A_1 X)_{i,j}$. We define the product nodes $(A_1)_{i,k} X_{k,j}$, the nodes corresponding to the sums $(A_1 X)_{i,j}$, and all intermediate nodes in the summation trees as *level-1 nodes*. Similarly, we define *level-2 nodes* as these nodes in the summation trees involved in computing $(A_1 X) A_2^\top$. We give an example of the summation tree with $d = 2$ in Fig. 3.

### 3.3 I/O COMPLEXITY

There are various ways to define the two-level memory hierarchy and the I/O complexity. We state the definition in Hong & Kung (1981), which formulates the two-level memory hierarchy as a red-blue pebble game played on a computational graph. Very recently, Saha & Ye (2024) proved that the I/O complexity of forward computation of FlashAttention is optimal by analyzing the red-blue pebble game on an attention forward computational graph.

**Definition 3.4** (Red-blue pebble game (Hong & Kung, 1981)). *Consider a game played on a directed acyclic graph that has a limited number of red pebbles and an unlimited number of blue pebbles. Initially, each input node (a node with no parents) is marked with a blue pebble, while all other nodes have no pebbles. The player is allowed to perform the following operations:*

- ***Input***: *Replace a blue pebble on a node with a red pebble.*

- ***Output***: *Replace a red pebble on a node with a blue pebble.*

- ***Compute***: *Place a red pebble on a node if all its parent nodes have red pebbles.*

- ***Delete***: *Remove a pebble from a node.*

*The objective of the game is to place blue pebbles on all output nodes (i.e., nodes with no children) while minimizing the total number of input and output operations used throughout the process.*

In the red-blue pebble game, each node represents a computational task. A red pebble denotes a unit in the small but fast layer known as *cache*, while a blue pebble represents a unit in the large but slower layer called *memory*. A task can only be computed once all its dependent tasks are completed. All computations are assumed to occur within the cache. Hence, efficient use of cache plays a critical role in reducing the I/O operations of an algorithm to minimize the cost associated with data transfer between memory and cache. We can define the I/O complexity by using the red-blue pebble game.

**Definition 3.5** (I/O complexity (Hong & Kung, 1981)). *Consider the red-blue pebble game played on a directed acyclic graph $G$. Let $M$ be a positive integer. The I/O complexity, denoted as $Q(G, M)$, is the minimum number of input and output operations to complete the objective of the game with the restriction that no more than $M$ red pebbles are present on the graph at any time. We omit $G$ when it is clear in the context.*

The red-blue pebble game provides insight into cache management by modeling the limited cache size through the number of red pebbles. The maximum number of red pebbles corresponds to the size of the cache, which means that there can be at most $M$ items in the cache at any given time.

## 4 MAIN RESULTS

In Theorem 1.1, we provide matching upper and lower bounds for the I/O complexity of attention gradient computation in the backward passes. In detail, Theorem 1.1 states that the I/O complexity of the attention gradient computation is $\Theta(\min\{\frac{n^2 d^2 + n d^3}{M}, \frac{n^2 d + n d^2}{\sqrt{M}}\})$, which splits the cache size into two cases: (1) small cache $M = o(d^2)$; (2) large cache $M = \Omega(d^2)$. At the cross point $M = d^2$, we have $\frac{n^2 d^2 + n d^3}{M} = \frac{n^2 d + n d^2}{\sqrt{M}} = n^2 + nd$. An intuitive figure of the asymptotic I/O complexity is shown in Fig. 1.

Here, we discuss two implications of Theorem 1.1. First, through the fine-grained analysis, our result identifies a critical point at $M = d^2$, where the I/O complexity changes its behavior. For $M = o(d^2)$, we establish better upper and lower bounds compared to existing results, demonstrating that FlashAttention is not optimal in this regime. Second, when $M = \Omega(d^2)$, Theorem 1.1 provides a tighter lower bound than existing work using red-blue pebble game (Definition 3.4), offering insights of algorithm design.

Moreover, by combining the results of Saha & Ye (2024) with our findings, we provide a more general and tighter I/O complexity characterization of FlashAttention 1/2 (Dao et al., 2022; Dao, 2023). In the large cache scenario where $M = \Omega(d^2)$, the attention forward I/O complexity is $\Theta\left(\frac{n^2 d^2}{M}\right)$, as discussed in Theorem 5.1 of Saha & Ye (2024). Combining this result with our attention backward I/O complexity $\Theta\left(\frac{n^2 d^2 + nd^3}{M}\right)$ (Theorem 1.1), we conclude that the overall complexity is $\Theta\left(\frac{n^2 d^2 + nd^3}{M}\right)$. Thus, given the cache size is sufficiently large, i.e., $M = \Omega(d^2)$, the I/O complexity of the forward and backward computation for FlashAttention 1/2 is optimal.

Our main result Theorem 1.1 is a summary of our results for different cache sizes (Theorem 4.1, 4.2, 4.3, and 4.4), which will be discussed in the later subsections.

## 4.1 Large Cache

The large cache scenario is more interesting and practical. We now prove an upper bound below.

**Theorem 4.1** (Large cache upper bound, informal version of Theorem D.5)**.** *Suppose $n$ is the input length, $d$ is the head dimension, and $M = \Omega(d^2)$ is the cache size. There is an algorithm (see Algorithm 9) outputs a $d \times d$ matrix $g = \frac{\mathrm{d}L(X)}{\mathrm{d}X}$ (Definition 3.2) with I/O complexity $O\left(\frac{n^2 d^2 + nd^3}{M}\right)$.*

We then demonstrate that this upper bound is tight by providing a matching lower bound for the I/O complexity of the attention backward passes. To achieve this, we employ the framework developed in Hong & Kung (1981), which shows that executing an algorithm on a machine with a two-level memory hierarchy can be modeled by a red-blue pebble game (Definition 3.4) on a directed acyclic graph. We present the large cache lower bound below, which shows as long as the cache size $M = \Omega(d^2)$, the I/O complexity is at least $\Omega\left(\frac{n^2 d^2 + nd^3}{M}\right)$.

**Theorem 4.2** (Large cache lower bound, informal version of Theorem E.9)**.** *Suppose $n$ is the input length and $d$ is the head dimension. Suppose the cache size $M = \Omega(d^2)$. Then the I/O complexity of attention gradient computation using standard matrix multiplication is always $\Omega\left(\frac{n^2 d^2 + nd^3}{M}\right)$.*

## 4.2 Small Cache

In the small cache case, we provide an upper bound below. Notice that this is better than the I/O complexity of FlashAttention since $O\left(\frac{n^2 d^2 + nd^3}{M}\right) > O\left(\frac{n^2 d + nd^2}{\sqrt{M}}\right)$ when $M = o(d^2)$.

**Theorem 4.3** (Small cache upper bound, informal version of Theorem C.12)**.** *Suppose $n$ is the input length, $d$ is the head dimension, and $M = o(d^2)$ is the cache size. There is an algorithm (see Algorithm 6) outputs a $d \times d$ matrix $g = \frac{\mathrm{d}L(X)}{\mathrm{d}X}$ (Definition 3.2) with I/O complexity $O\left(\frac{n^2 d + nd^2}{\sqrt{M}}\right)$, time complexity $O(n^2 d + nd^2)$, and space complexity $O(n^2 + d^2)$.*

Furthermore, we show that attention gradient computation can be reduced to matrix multiplication, establishing a matching lower bound.

**Theorem 4.4** (Small cache lower bound, informal version of Theorem E.10)**.** *Suppose $n$ is the input length and $d$ is the head dimension. Suppose the cache size $M = o(d^2)$. Then the I/O complexity of attention gradient computation using standard matrix multiplication is always $\Omega\left(\frac{n^2 d + nd^2}{\sqrt{M}}\right)$.*

Our theory suggests that, when the commonly used hidden dimension size increases while some commercial GPU cache sizes remain insufficiently large, our algorithm designed for small cache sizes would become relevant and useful. For example, the current network architectures usually set $d = 128$, so the dividing point is approximately $d^2 \times$ size_of(data type), e.g., $16,384 \times 32$-bit = 65.5 KB for float32. For the NVIDIA A100 GPU, the size of each streaming multiprocessor (SM/L1 cache) is 192KB, so we can choose FlashAttention. However, for old GPUs such as

NVIDIA GTX1060, the size of each SM is 48 KB, so the algorithm for the small cache size is suitable. Whether a cache is large or small depends on the relation between $M$ and $d$, not the absolute size. We can also view the small-cache case as a high-dimensional scenario, which may apply to other settings. Hence, our work provides theoretical insights and could guide future developments in attention mechanisms tailored to evolving hardware limitations.

### 4.3 LOWER BOUND OF SPARSE ATTENTION FORWARD AND BACKWARD PASSES

Sparse attention is a generalization of standard attention and has been popular in practical applications. We refer readers to Section 2 for more discussion. To state our results, we first introduce some notations. For any matrix $A$, we use $\mathrm{nnz}(A)$ to denote the number of non-zero entries in the matrix $A$. We assume that sparse matrices are stored by listing only their non-zero entries along with their coordinates. We assume sparse semi-ring matrix multiplication, which restricts operations to the addition and multiplication of these entries. Each output entry $(AB)_{i,j}$ can only be computed as the sum of products given by $\sum_k A_{i,k} B_{k,j}$.

**Theorem 4.5** (Lower bound for sparse attention forward and backward, formal version of Theorem 1.2). *Suppose $n$ is the input length, $d$ is the head dimension, and $M$ is the cache size. Let $Z_A := \min\{\mathrm{nnz}(A_1), \mathrm{nnz}(A_2)\}, Z_X := \mathrm{nnz}(X), Z_{AX} = \min\{\mathrm{nnz}(A_1 X), \mathrm{nnz}(X A_2^\top)\}, Z_{AXA} := \mathrm{nnz}(A_1 X A_2^\top)$. Then any algorithm for both attention forward and backward computation using sparse semi-ring matrix multiplication has I/O complexity*

$$\Omega\left(\min\left\{\frac{Z_A^2 + Z_A Z_X}{M}, \frac{Z_A\sqrt{Z_{AXA}} + \sqrt{Z_A Z_X Z_{AX}}}{\sqrt{M}}\right\}\right).$$

**Remark 4.6.** *When matrices involved in attention computation are dense, i.e., $Z_A = \Omega(nd), Z_X = \Omega(d^2), Z_{AX} = \Omega(nd)$, and $Z_{AXA} = \Omega(n^2)$. In such case, our lower bound reads as $\Omega(\min\{\frac{n^2 d^2 + nd^3}{M}, \frac{n^2 d + nd^2}{\sqrt{M}}\})$. Hence, it matches the result of lower bounds in the dense case.*

**Remark 4.7** (The dividing point for sparse attention). *The dividing point of small cache and large cache can be computed by equaling two lower bounds, i.e., $\frac{Z_A^2 + Z_A Z_X}{M} = \frac{Z_A\sqrt{Z_{AXA}} + \sqrt{Z_A Z_X Z_{AX}}}{\sqrt{M}}$. Rearranging the equation gives $\sqrt{M} = \frac{Z_A^2 + Z_A Z_X}{Z_A\sqrt{Z_{AXA}} + \sqrt{Z_A Z_X Z_{AX}}}$. Note that when matrices are dense, we have $\sqrt{M} = \frac{n^2 d^2 + nd^3}{n^2 d + nd^2} = \frac{d + d^2/n}{1 + d/n}$. Since we assume that $n \gg d$, this is exactly $\sqrt{M} = d$, i.e., $M = d^2$, which matches the dividing point of the dense case.*

## 5 TECHNICAL OVERVIEW

**Upper Bound of Small Cache.** In Section C, we present algorithms for the backward passes of attention in the small cache case, where $M = o(d^2)$. We observe that when $M = o(d^2)$, we have $\frac{n^2 d^2 + nd^3}{M} > \frac{n^2 d + nd^2}{\sqrt{M}} > n^2 + nd$. Then we can exploit this to design a better algorithm with I/O complexity better than $\frac{n^2 d^2 + nd^3}{M}$, by reading/writing the $n \times n$ attention matrix and other $n \times d$ intermediate matrices from/to memory. In detail, our small cache algorithm (Algorithm 6) follows the computational graph in Figure 2 and is divided into four phases. In Phase 1 (Algorithm 2), we compute the attention matrix $f$ (Definition B.5) and write it to memory. In Phase 2 (Algorithm 3), we compute $q$ (Definition B.8), incorporating the information from the upstream gradient $\mathrm{d}O$. Phase 3 (Algorithm 4) computes the gradient component matrix $p$ (Definition B.9). Finally, in Phase 4 (Algorithm 5), we compute the final gradient $g = A_1^\top p A_2$ (Definition 3.2). At a high level, our algorithm splits the input and output matrices into blocks of size $\sqrt{M} \times \sqrt{M}$. On the other hand, FlashAttention divides the $n \times d$ input matrices into multiple $k \times d$ matrices, where $k < n$. Compared to our upper bound, we can see that FlashAttention is not optimal in this case. Following the computational graph in Figure 2, we perform the backward passes of attention using each $\sqrt{M} \times \sqrt{M}$ block as basic elements in standard matrix multiplication. Compared to forward passes, the computational graph of backward passes is more complicated and requires more fine-grained analysis, e.g., the four phases mentioned above. Through a detailed analysis of Algorithm 6, we establish Theorem 4.3.

**Upper Bound of Large Cache.** In Section D, we present algorithms for attention backward in the large cache case, where $M = \Omega(d^2)$. Similar to FlashAttention, the $n \times n$ attention matrix

$f$ (Definition B.5) cannot be directly loaded into the cache, even though it has been computed and can be stored in memory. The overall algorithm (Algorithm 9) consists of two phases. In Phase 1 (Algorithm 7), we compute $S = A_1 X$ and $h = A_3 Y$, and these two matrices are then passed to Phase 2. In Phase 2 (Algorithm 8), the inputs are matrices $A_1, A_2, S, h, O, \mathrm{d}O \in \mathbb{R}^{n \times d}$ (Definitions 3.1, B.6, B.7, and B.8), and vector $l \in \mathbb{R}^n$ (Definition B.4). We vertically divide the inputs into row block matrices of size $B_r \times d$ or $B_c \times d$, where $B_r = \min\{\lceil M/4d \rceil, d\}$ and $B_c = \lceil M/4d \rceil$. Using these row block matrices as computation units, we follow the computational graph (Fig. 2) and FlashAttention's procedure. After accounting for the reads and writes of the overall algorithm (Algorithm 9), we prove Theorem 4.1. When the cache size is as large as $\Theta(nd)$, the I/O complexity can be reduced to $O(nd + d^2)$, which corresponds to the size of the input and output of the algorithm.

**Lower Bound of Large Cache and Small Cache.** In Section E, we establish the lower bounds for the I/O complexity of attention gradient computation in both large and small cache cases. Following Definitions 3.4 and 3.5, we analyze the red-blue pebble game on the computational graph of any attention backward algorithm using standard matrix multiplication. More specifically, the key concept is the $M$-partition, which decomposes the graph into subgraphs, ensuring that each subgraph satisfies conditions related to dominator and minimum sets (Definitions E.1, E.2, E.3, E.4, and E.5). Our proofs for the lower bound of backward passes builds upon the lemmas (Lemmas E.7 and E.8), which provide the foundation for relating the number of subgraphs to the I/O operations required. For the large cache scenario, $M = \Omega(d^2)$, we demonstrate that the I/O complexity scales with the need to compute matrix products efficiently. In the small cache case, $M = o(d^2)$, we show that higher I/O complexity is unavoidable due to the data transfers between cache and memory by reducing to the standard matrix multiplication. These analyses are formally established in the proofs of Theorems E.9 and E.10. Our Theorems E.10, the small cache lower bound case, requires a new analysis of deviation.

**Remark 5.1.** *The Softmax in Definition 3.1 can be changed to other non-linear activation functions and our lower bound still holds. It is because we must compute matrix multiplication of size $n \times d$ and $d \times n$ in non-linear attention. However, for linear attention, that is, $A_1 X A_2^\top A_3 Y$, our lower bound is loose. This is because we can compute $\underbrace{A_2^\top}_{d \times n} \underbrace{A_3}_{n \times d}$ first, and then compute $\underbrace{A_1}_{n \times d} \underbrace{X}_{d \times d} \underbrace{A_2^\top A_3}_{d \times d} \underbrace{Y}_{d \times d}$.*

**Lower Bound of Sparse Attention Forward and Backward Passes.** In Section F, we establish lower bounds on the I/O complexity of sparse attention computation for both forward and backward passes. Sparse matrix multiplication is considered, where only non-zero entries are stored and used in computations. We derive I/O complexity bounds based on the non-zero counts of input matrices and the I/O operations required for sparse matrix multiplication (Lemma F.1). We extend these bounds to the matrix products involved in the attention mechanism (Lemma F.2), which requires multiple sparse matrix multiplication analysis. We analyze scenarios where matrices are stored in cache or require intermediate I/Os during computation to obtain the I/O complexity bounds for both forward and backward passes (Theorems F.3 and F.4), and Theorem 4.5 directly holds as a consequence.

# 6 CONCLUSION

In this work, we established tight bounds on I/O complexity for both small and large caches. Our results confirm that FlashAttention is optimal for both forward and backward on large cache sizes. For small cache sizes, we provided improved upper and lower bounds compared to existing methods. Additionally, we derived lower bounds for sparse attention for both forward and backward and across cache sizes. Our findings complete the theoretical foundation for I/O complexity in attention mechanisms and provide a deeper understanding of memory efficiency in attention computations, offering the insights for optimizing implementations in future deep learning architecture, and speeding up training and inference of large language models.

## ETHIC STATEMENT

This paper does not involve human subjects, personally identifiable data, or sensitive applications. We do not foresee direct ethical risks. We follow the ICLR Code of Ethics and affirm that all aspects of this research comply with the principles of fairness, transparency, and integrity.

## REPRODUCIBILITY STATEMENT

We ensure reproducibility of our theoretical results by including all formal assumptions, definitions, and complete proofs in the appendix. The main text states each theorem clearly and refers to the detailed proofs. No external data or software is required.

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

# Appendix

**Roadmap.** In Section A, we present a more comprehensive overview of related work pertinent to our study. In Section B, we introduce additional preliminaries, including notations and definitions of intermediate variables. Section C provides algorithms and establishes an upper bound theorem for the attention backward pass in small cache case $M = o(d^2)$. In Section D, we offer algorithms and an upper bound theorem for the attention backward pass in large cache case $M = \Omega(d^2)$. In Section E, we provide proofs for our attention backward I/O complexity lower bound results. In Section F, we prove the I/O complexity lower bounds for sparse attention. In Section G, we discuss the social impact of our work.

## A  MORE RELATED WORK

**Large Language Models.** The exceptional success of generative large language models (LLMs), such as GPT-4 (Achiam et al., 2023), Claude 3 (Anthropic, 2024), Gemini 1.5 (Reid et al., 2024), Llama 3.1 (Llama Team, 2024), Mistral Nemo (Jiang et al., 2023), Phi 3.5 (Abdin et al., 2024), is fundamentally attributed to the transformer architecture introduced by Vaswani et al. (2017) and all support at least 128k input token length. The transformer architecture and its self-attention mechanism have become indispensable in leading natural language processing (NLP) models (Chang et al., 2024), demonstrating remarkable capabilities across a diverse array of applications, including language translation (He et al., 2021), sentiment analysis (Usama et al., 2020), language modeling (Martin et al., 2019), the integration of differential privacy (Singh et al., 2024; Liang et al., 2024e), and multi-modal tasks (Zhang et al., 2024a; Liang et al., 2024f; Wang et al., 2024). Transformers' emergent compositional abilities (Dziri et al., 2024; Xu et al., 2024b) and proficiency in in-context learning (Olsson et al., 2022; Min et al., 2022; Shi et al., 2024b) have led some to consider them as early indicators of Artificial General Intelligence (AGI) (Bubeck et al., 2023). As such, the transformer architecture plays a pivotal role in advancing the field of AI.

**Attention Computation Acceleration.** The quadratic time complexity of attention computation with respect to the length of the input sequence (Vaswani et al., 2017) poses significant computational challenges, especially for long sequences. Consequently, accelerating attention computation has become a crucial research area. From a theoretical standpoint, numerous works focus on approximating the attention matrix to accelerate computation (Han et al., 2024; Alman & Song, 2023; 2024a; Liang et al., 2024c; Alman & Song, 2024b; Liang et al., 2024f). Experimental approaches involve modifying model architectures and optimizing implementations to accelerate inference. Methods such as Mamba (Gu & Dao, 2023; Dao & Gu, 2024), Linearizing Transformers (Zhang et al., 2024b; Mercat et al., 2024), Hopfield Models (Hu et al., 2023; Wu et al., 2024b; Hu et al., 2024c; Xu et al., 2024a; Wu et al., 2024a; Hu et al., 2024a;b) and PolySketchFormer (Zandieh et al., 2023; Kacham et al., 2023) aim to improve model performance and inference speed. System-level optimizations, such as FlashAttention (Dao et al., 2022; Dao, 2023; Shah et al., 2024) and block-wise parallel decoding (Stern et al., 2018), address bottlenecks in attention mechanisms and enhance inference speed through efficient implementation strategies. Collectively, these advancements contribute to making attention mechanisms more scalable and efficient, facilitating the deployment of large-scale language models. (Shi et al., 2024a) accelerates inference by compressing the input text.

**More about Attention Computation Acceleration.** The quadratic time complexity of attention computation with respect to the length of the input sequence (Vaswani et al., 2017) poses significant computational challenges, especially for long sequences. Consequently, accelerating attention computation has become a crucial research area, with approaches broadly divided into two categories: (1) theoretical optimization of computational complexity (Alman & Song, 2023; 2024a), and (2) experimental improvements to model performance (Dao et al., 2022; Dao, 2023; Shah et al., 2024; Ge et al., 2023; Feng et al., 2024a).

From a theoretical standpoint, numerous works focus on approximating the attention matrix to accelerate computation. For example, Alman & Song (2023; 2024a) utilize polynomial kernel approximation techniques (Aggarwal & Alman, 2022) to speed up both training and inference of a single attention layer, achieving almost linear time complexity, and extend this approach to multi-layer transformer (Liang et al., 2024c) and tensor attention (Alman & Song, 2024b; Liang et al., 2024f). Other theoretical contributions include the conv-basis method introduced by Liang et al. (2024a)

and a near-linear time algorithm proposed by Han et al. (2024) under the assumptions of uniform softmax column norms and sparsity.

Experimental approaches involve modifying model architectures and optimizing implementations to accelerate inference. Methods such as Mamba (Gu & Dao, 2023; Dao & Gu, 2024), Linearizing Transformers (Zhang et al., 2024b; Mercat et al., 2024), PolySketchFormer (Zandieh et al., 2023; Kacham et al., 2023), and various implementations of the Hopfield Model (Hu et al., 2024b;a; Wu et al., 2024a; Xu et al., 2024a; Hu et al., 2024c; Wu et al., 2024b; Hu et al., 2023) aim to improve model performance and inference speed. Additionally, specific techniques like weight pruning (Liang et al., 2024b; Li et al., 2024) have been developed to accelerate LLM generation. Some other techniques are introduced for efficient adaptation, such as LoRA (Hu et al., 2022; Zeng & Lee, 2024; Hu et al., 2024d) and prefix turning (Li & Liang, 2021; Liang et al., 2024d). System-level optimizations, such as Flash Attention (Dao et al., 2022; Dao, 2023; Shah et al., 2024) and block-wise parallel decoding (Stern et al., 2018), address bottlenecks in attention mechanisms and enhance inference speed through efficient implementation strategies. Collectively, these advancements contribute to making attention mechanisms more scalable and efficient, facilitating the deployment of large-scale language models.

**More about Learning with Bounded Memory and I/O Complexity.** Learning with bounded memory has been studied in various fields in machine learning such as online learning (Maiti et al., 2021; Srinivas et al., 2022; Peng & Rubinstein, 2023; Peng & Zhang, 2023), parity learning (Steinhardt et al., 2016; Raz, 2017; 2018; Garg et al., 2018), convex optimization (Woodworth & Srebro, 2019; Marsden et al., 2022; Chen & Peng, 2023), active learning (Hopkins et al., 2021), learning linear classifiers (Brown et al., 2022), attention computation (Addanki et al., 2023), linear regression (Steinhardt & Duchi, 2015; Sharan et al., 2019; Brown et al., 2022), linear programming (Tauman Kalai et al., 2016; Liu et al., 2020), semi-definite programming (Song et al., 2023), principal component analysis (Deng et al., 2023), continual learning (Chen et al., 2022; Ermis et al., 2022), entropy estimation (Acharya et al., 2019; Aliakbarpour et al., 2022) and others (Moshkovitz & Tishby, 2017; Gonen et al., 2020).

A common memory model in computational systems is the two-level memory hierarchy. In this model, there are two layers of memory: a small but fast layer called the *cache*, and a large but slower layer called the *memory*. The I/O (input/output) complexity of an algorithm measures its efficiency based on the number of data transfer operations it performs between the cache and the memory. In domains such as big data analytics and database management, these data transfers can become significant performance bottlenecks because massive datasets cannot be entirely accommodated in the cache, and thus optimizing I/O is essential for fast data retrieval and storage, directly impacting query performance and system scalability (Gropp et al., 2014; Zhang et al., 2015). The early work of Hong & Kung (1981) formulated the I/O complexity mathematically using the language of graph theory. Vitter (2001) provides a comprehensive survey of the I/O complexity of various batched and online problems. There exists a substantial body of work on the I/O complexity of numerous problems, including sorting (Aggarwal & Vitter, 1988), graph algorithms (Cui et al., 2020; Jain & Zaharia, 2020; Jiang et al., 2021; Deng & Tao, 2024), fine-grained I/O complexity (Demaine et al., 2017), computational trade-off in data transfers (Demaine & Liu, 2018), computing prime tables (Bender et al., 2016), attention computation (Saha & Ye, 2024), integer multiplication (Bilardi & De Stefani, 2019; De Stefani, 2019b), and matrix multiplication (De Stefani, 2019a; Nissim & Schwartz, 2019).

**Sparse Attention.** Over the past few years, there has been extensive research on sparse Transformer/Attention models with weights pruning and inputs pruning, aimed at accelerating computation and training (Ye et al., 2019; Sukhbaatar et al., 2019; Beltagy et al., 2020; Tay et al., 2020; Guo et al., 2023; Shirzad et al., 2023; Sun et al., 2024; Li et al., 2024; Deng et al., 2024; Chen et al., 2024a). In practice, the attention matrix is sparse, significantly reducing computational costs. Theoretical studies, such as Yun et al. (2020), have demonstrated that sparse transformers are expressive enough and can achieve universal approximation properties.

# B PRELIMINARY

In Section B.1, we define some basic notation we will use. In Section B.2, we introduce the memory hierarchy we consider. In Section B.3, we state important facts related to fast matrix multiplication. In Section B.4, we define several intermediate functions which will arise in our algorithms.

## B.1 NOTATIONS

For any positive integer $n$, we define $[n] := \{1, 2, \ldots, n\}$. For two same length vector $x$ and $y$, we use $\langle x, y \rangle$ to denote the inner product between $x$ and $y$, i.e., $\langle x, y \rangle = \sum_{i=1}^{n} x_i y_i$. We use $\circ$ to denote the Hadamard product i.e. the $(i, j)$-entry of $A \circ B$ is $A_{i,j} B_{i,j}$. We use $x \circ y$ to denote vector that $i$-th entry is $x_i y_i$. Let $\mathbf{1}_n$ denote the length-$n$ all ones vector. It is not hard to see that $\langle x \circ y, \mathbf{1}_n \rangle = \langle x, y \rangle$. For a vector $x$, we use $x^\top$ to denote the transpose of $x$. For a matrix $A$, we use $A^\top$ to denote the transpose of matrix $A$. For a matrix $A$, we use $\exp(A)$ to denote the matrix that $(i, j)$-th coordinate is $\exp(A_{i,j})$.

Given a matrix $A \in \mathbb{R}^{n \times m}$, we index an individual entry as $A[i, j]$. The $i$-th row is denoted $A[i]$ while the $j$-th column is denoted $A[*, j]$. $A[i_1 : i_2, j_1 : j_2]$ denotes a block of $A$ consisting of entries $(i, j)$ where $i \in [i_1, i_2]$ and $j \in [j_1, j_2]$. Given a block size $B$, the block $A[(i-1) \cdot B + 1 : i \cdot B, (j-1) \cdot B + 1 : j \cdot B]$ is denoted $A^{(B)}[i, j]$.

For a vector $v \in \mathbb{R}^n$, we similarly denote entries $v[i]$, a contiguous block of entries as $v[i_1 : i_2]$, and the $i$-th block of size $B$ as $v^{(B)}[i]$. Let $\mathrm{diag}(v)$ denote the matrix $D \in \mathbb{R}^{n \times n}$ with $D[i, i] = v[i]$.

## B.2 MEMORY HIERARCHY

In this study, we consider a two-level memory hierarchy composed of a small but fast layer called the *cache* and a large, slower layer referred to as the *memory*. We assume that the memory has unlimited capacity, while the cache is constrained by a finite size $M$. Moreover, all computations are performed exclusively within the cache.

## B.3 MATRIX MULTIPLICATION

We define matrix multiplication notation and state some well-known facts here.

**Definition B.1.** *Let $n_1, n_2, n_3$, denote any three positive integers. We use $\mathcal{T}_{\mathrm{mat}}(n_1, n_2, n_3)$ to denote the time of multiplying an $n_1 \times n_2$ matrix with another $n_2 \times n_3$.*

Then, we introduce a well-known fact.

**Fact B.2.** *Let $n_1, n_2, n_3$, denote any three positive integers. $\mathcal{T}_{\mathrm{mat}}(n_1, n_2, n_3) = O(\mathcal{T}_{\mathrm{mat}}(n_1, n_3, n_2)) = O(\mathcal{T}_{\mathrm{mat}}(n_2, n_1, n_3)) = O(\mathcal{T}_{\mathrm{mat}}(n_2, n_3, n_1)) = O(\mathcal{T}_{\mathrm{mat}}(n_3, n_1, n_2)) = O(\mathcal{T}_{\mathrm{mat}}(n_3, n_2, n_1))$.*

## B.4 DEFINITIONS OF INTERMEDIATE VARIABLES

We start by some definitions about $X \in \mathbb{R}^{d \times d}$.

**Definition B.3** (Definition 3.4 in Alman & Song (2024a)). *Let $A_1, A_2 \in \mathbb{R}^{n \times d}$ be two matrices. Let $X \in \mathbb{R}^{d \times d}$.*

*Let us define function $A(X)$ to be:*

$$A(X) := \underbrace{\exp(A_1 X A_2^\top)}_{n \times n}.$$

**Definition B.4** (Definition 3.5 in Alman & Song (2024a)). *For $A(X) \in \mathbb{R}^{n \times n}$ defined in Definition B.3, we define the softmax normalizing vector $l(X) \in \mathbb{R}^n$ to be*

$$l(X) := \underbrace{A(X)}_{n \times n} \cdot \underbrace{\mathbf{1}_n}_{n \times 1}.$$

**Definition B.5** (Definition 3.6 in Alman & Song (2024a)). *Suppose that $l(X) \in \mathbb{R}^n$ is defined as in Definition B.4. Let $A(X) \in \mathbb{R}^{n \times n}$ be defined as in Definition B.3. For a fixed $j_0 \in [n]$, let us consider $f(X)_{j_0}$*

$$f(X)_{j_0} := \underbrace{l(X)_{j_0}^{-1}}_{\text{scalar}} \underbrace{A(X)_{j_0}}_{n \times 1}.$$

*Let $f(X) \in \mathbb{R}^{n \times n}$ denote the matrix where $j_0$-th row is $(f(X)_{j_0})^\top$.*

*Furthermore, the matrix form of $f(X)$ is*

$$f(X) = \text{diag}(l(X))A(X)$$

*We then define $h(Y)$ related to $Y \in \mathbb{R}^{d \times d}$.*

**Definition B.6** (Definition 3.7 in Alman & Song (2024a)). *For $A_3 \in \mathbb{R}^{n \times d}$ and $Y \in \mathbb{R}^{d \times d}$, we define $h(Y) \in \mathbb{R}^{n \times d}$ as:*

$$h(Y) := \underbrace{A_3}_{n \times d} \underbrace{Y}_{d \times d}.$$

*Let us define the forward output matrix $O$.*

**Definition B.7.** *Let $f(X), h(Y)$ be defined in Definition B.5 and B.6. We define the output of attention as:*

$$O := \underbrace{f(X)}_{n \times n} \underbrace{h(Y)}_{n \times d}$$

*where $O \in \mathbb{R}^{n \times d}$ is the output matrix of attention forward computation.*

*Now, we define $q$, which incorporates the information from upstream gradient.*

**Definition B.8** (Definition C.10 in Liang et al. (2024c)). *Let $\mathrm{d}O \in \mathbb{R}^{n \times d}$ be the upstream gradient, the matrix resulting from the application of the chain rule. Define $h(Y) \in \mathbb{R}^{n \times d}$ as in Definition B.6.*

*We define $q(Y) \in \mathbb{R}^{n \times n}$ as*

$$q(Y) := \underbrace{\mathrm{d}O}_{n \times d} \underbrace{h(Y)^\top}_{d \times n}$$

*Then we use $q(Y)_{j_0}^\top$ to denote the $j_0$-th row of $q(Y) \in \mathbb{R}^{n \times n}$.*

*Finally, we define the gradient component matrix $p$.*

**Definition B.9** (Definition C.5 in Alman & Song (2024a)). *For every index $j_0 \in [n]$, we define $p(X)_{j_0} \in \mathbb{R}^n$ as*

$$p(X)_{j_0} := (\text{diag}(f(X)_{j_0}) - f(X)_{j_0}f(X)_{j_0}^\top)q(Y)_{j_0}.$$

*We define $p(X) \in \mathbb{R}^{n \times n}$ in the sense that $p(X)_{j_0}^\top$ is the $j_0$-th row of $p(X)$. Additionally, $p(X)$ has matrix form as*

$$p(X) = f(X) \circ q(Y) - \text{diag}((f(X) \circ q(Y)) \cdot \mathbf{1}_n)f(X)$$
$$= f(X) \circ q(Y) - \text{diag}((O \circ \mathrm{d}O) \cdot \mathbf{1}_n)f(X)$$

*where $f(X), O$ are defined in Definition B.5 and B.7, and $q(Y), \mathrm{d}O$ are defined in Definition B.8.*

## C  I/O COMPLEXITY UPPER BOUND FOR SMALL CACHE

In this section, we prove the I/O complexity upper bound (Theorem C.12) for small cache case $M = o(d^2)$. Specifically, in Section C.1, we introduce an algorithm of attention gradient computation without cache to guide our algorithm design. Section C.2 presents algorithms and analyses for attention gradient computation in the small cache setting. Finally, Section C.3 provides the upper bound theorem for the small cache case.

## C.1 Algorithm for Attention Backward Without Cache

Using results from Alman & Song (2024a), we can compute the gradient in $\mathcal{T}_{\mathrm{mat}}(n, d, n) + \mathcal{T}_{\mathrm{mat}}(n, d, d)$ time.

**Lemma C.1** (Attention gradient computation, Lemma C.8 in Alman & Song (2024a)). *If it holds that*

- *Define $A_1, A_2, A_3, \mathrm{d}O \in \mathbb{R}^{n \times d}$. Define $X, Y \in \mathbb{R}^{d \times d}$ to be several input fixed matrices.*

- *Let $X, Y \in \mathbb{R}^{d \times d}$ denote matrix variables (we will compute gradient with respect to $X$).*

- *Let $g = \frac{\mathrm{d}L(X)}{\mathrm{d}X} \in \mathbb{R}^{d \times d}$ (Definition 3.2).*

*Then, gradient $g \in \mathbb{R}^{d \times d}$ can be computed in $\mathcal{T}_{\mathrm{mat}}(n, d, n) + \mathcal{T}_{\mathrm{mat}}(n, d, d)$ time.*

We first give a naive algorithm that have not utilized cache to compute the gradient (Algorithm 1).

---

**Algorithm 1** Attention gradient computation without cache. See more details in Section B and C of Alman & Song (2024a) and Section F of Liang et al. (2024c).

---

1: **procedure** ATTENTIONGRADIENTNOCACHE($A_1, A_2, A_3, \mathrm{d}O \in \mathbb{R}^{n \times d}, X, Y \in \mathbb{R}^{d \times d}$)    ▷ Lemma C.2, Lemma C.3
2:     Read $A_1, A_2, X$, initialize $A \leftarrow 0^{n \times n}$, compute $A \leftarrow A + A_1 X A_2^\top$, and delete $X$
3:     Compute $A \leftarrow \exp(A)$, initialize $l \leftarrow 0^n$, and compute $l \leftarrow l + A \cdot \mathbf{1}$
4:     Initialize $f \leftarrow 0^{n \times n}$, compute $f \leftarrow f + \mathrm{diag}(l)^{-1} A$, and delete $A, d$
5:     Read $A_3, Y$, initialize $h \leftarrow 0^{n \times d}$, compute $h \leftarrow h + A_3 Y$, and delete $A_3, Y$
6:     Read $\mathrm{d}O$, initialize $q \leftarrow 0^{n \times n}$, compute $q \leftarrow q + \mathrm{d}O h^\top$, and delete $\mathrm{d}O, h$
7:     Initialize $p \leftarrow 0^{n \times n}$, compute $p \leftarrow p + f \circ q - \mathrm{diag}((f \circ q) \cdot \mathbf{1}) f$, and delete $f, q$
8:     Initialize $g \leftarrow 0^{n \times n}$, compute $g \leftarrow g + A_1^\top p A_2$, and delete $A_1, A_2, p$
9:    **return** $g$         ▷ $g = \frac{\mathrm{d}L(X)}{\mathrm{d}X} \in \mathbb{R}^{d \times d}$, see Definition 3.2
10: **end procedure**

---

**Lemma C.2** (Correctness). *The ATTENTIONGRADIENTNOCACHE (Algorithm 1) outputs a $d \times d$ matrix $\frac{\mathrm{d}L(X)}{\mathrm{d}X}$ defined in Definition 3.2.*

*Proof.* From Lemma C.1, we know this holds. $\square$

**Lemma C.3** (Time/space complexity). *There exists an algorithm (see Algorithm 1) that can compute the exact gradient in Definition 3.2 in $\mathcal{T}_{\mathrm{mat}}(n, d, n) + \mathcal{T}_{\mathrm{mat}}(n, d, d)$ time and $O(n^2 + d^2)$ space.*

*Proof.* From Lemma C.1, we can prove the time complexity. Since the stored matrices have three sizes, namely $n \times d, n \times n, d \times d$, the space complexity is $O(n^2 + nd + d^2) = O(n^2 + d^2)$. $\square$

## C.2 Algorithms for Attention Backward in Small Cache

We now give algorithms to compute the upper bound of small cache case $M = o(d^2)$ in attention backward computation.

First, we give the algorithm and analysis for Phase 1 (see Algorithm 2) to compute $f$ defined in Definition B.5.

**Lemma C.4** (Correctness of Phase 1). *The ATTENTIONGRADIENTCACHEPHASE1 (Algorithm 2) outputs a $n \times n$ matrix $f$ defined in Definition B.5.*

*Proof.* The algorithm first computes $S = A_1 X$. Then it computes $A = S A_2^\top$, $A = \exp(A)$, and $l = A \cdot \mathbf{1}$. Finally, it outputs $f = \mathrm{diag}(l)^{-1} A$ which is $f$ defined in Definition B.5. $\square$

**Lemma C.5** (I/O complexity of Phase 1). *The I/O complexity of ATTENTIONGRADIENTCACHEPHASE1 (Algorithm 2) is $O(\frac{n^2 d + n d^2}{\sqrt{M}})$.*

*Proof.* In Phase 1 (Algorithm 2) the number of items in cache is at most $3B^2 + B \leq 4B^2 \leq M$. For each iteration in computing $S = A_1X$ and $A = SA_2^\top$, the algorithm reads $O(B^2)$ from memory into cache. This is the dominating factor of the I/O complexity of the algorithm. Thus, the I/O complexity of Phase 1 is $O(\frac{n^2d}{B^3}B^2) + O(\frac{nd^2}{B^3}B^2) = O(\frac{n^2d+nd^2}{B}) = O(\frac{n^2d+nd^2}{\sqrt{M}})$. $\qquad\square$

---

**Algorithm 2** Attention gradient computation with cache phase 1. Compute $f$.

---

1: **procedure** ATTENTIONGRADIENTCACHEPHASE1($A_1, A_2 \in \mathbb{R}^{n \times d}, X \in \mathbb{R}^{d \times d}, M \in \mathbb{N}_+$) $\triangleright$ Lemma C.4, Lemma C.5
2: $\quad B \leftarrow \lfloor\sqrt{M/4}\rfloor$
3: $\quad$ /*Phase 1: Compute $f$*/
4: $\quad$ **for** $1 \leq i \leq \lceil n/B \rceil$ **do**
5: $\quad\quad$ **for** $1 \leq j \leq \lceil d/B \rceil$ **do**
6: $\quad\quad\quad$ Initialize $S^{(B)}[i,j] \leftarrow 0^{B \times B}$ in cache
7: $\quad\quad\quad$ **for** $1 \leq k \leq \lceil d/B \rceil$ **do**
8: $\quad\quad\quad\quad$ Read $A_1^{(B)}[i,k]$ and $X^{(B)}[k,j]$ into cache
9: $\quad\quad\quad\quad$ Compute $S^{(B)}[i,j] \leftarrow S^{(B)}[i,j] + A_1^{(B)}[i,k]X^{(B)}[k,j]$ in cache $\quad \triangleright S = A_1X$
10: $\quad\quad\quad\quad$ Delete $A_1^{(B)}[i,k]$ and $X^{(B)}[k,j]$ from cache
11: $\quad\quad\quad$ **end for**
12: $\quad\quad\quad$ Write $S^{(B)}[i,j]$ in to memory, and delete $S^{(B)}[i,j]$ from cache
13: $\quad\quad$ **end for**
14: $\quad$ **end for**
15: $\quad$ **for** $1 \leq i \leq \lceil n/B \rceil$ **do**
16: $\quad\quad$ Initialize $l^{(B)}[i] \leftarrow 0^B$ in cache
17: $\quad\quad$ **for** $1 \leq j \leq \lceil n/B \rceil$ **do**
18: $\quad\quad\quad$ Initialize $A^{(B)}[i,j] \leftarrow 0^{B \times B}$ in cache
19: $\quad\quad\quad$ **for** $1 \leq k \leq \lceil d/B \rceil$ **do**
20: $\quad\quad\quad\quad$ Read $S^{(B)}[i,k]$ and $(A_2^\top)^{(B)}[k,j]$ into cache
21: $\quad\quad\quad\quad$ Compute $A^{(B)}[i,j] \leftarrow A^{(B)}[i,j] + S^{(B)}[i,k](A_2^\top)^{(B)}[k,j]$ in cache $\qquad\qquad \triangleright$
$\quad\quad\quad A = SA_2^\top$
22: $\quad\quad\quad\quad$ Delete $S^{(B)}[i,k]$ and $(A_2^\top)^{(B)}[k,j]$ from cache
23: $\quad\quad\quad$ **end for**
24: $\quad\quad\quad$ Compute $A^{(B)}[i,j] \leftarrow \exp(A^{(B)}[i,j])$ in cache, and write $A^{(B)}[i,j]$ into memory
25: $\quad\quad\quad$ Compute $l^{(B)}[i] \leftarrow l^{(B)}[i] + A^{(B)}[i,j] \cdot \mathbf{1}$ in cache $\qquad\qquad \triangleright l = A \cdot \mathbf{1}$
26: $\quad\quad\quad$ Delete $A^{(B)}[i,j]$ from cache
27: $\quad\quad$ **end for**
28: $\quad\quad$ **for** $1 \leq j \leq \lceil n/B \rceil$ **do**
29: $\quad\quad\quad$ Initialize $f^{(B)}[i,j] \leftarrow 0^{B \times B}$ in cache
30: $\quad\quad\quad$ Read $A^{(B)}[i,j]$ into cache
31: $\quad\quad\quad$ Compute $f^{(B)}[i,j] \leftarrow f^{(B)}[i,j] + \text{diag}(l^{(B)}[i])^{-1}A^{(B)}[i,j]$
32: $\quad\quad\quad$ Write $f^{(B)}[i,j]$ into memory, and delete $A^{(B)}[i,j]$ and $f^{(B)}[i,j]$ from cache
33: $\quad\quad$ **end for**
34: $\quad\quad$ Delete $l^{(B)}[i]$ from cache
35: $\quad$ **end for**
36: $\quad$ **return** $f$ $\qquad\qquad\qquad \triangleright f \in \mathbb{R}^{n \times n}$, where $f$ is defined in Definition B.5
37: **end procedure**

---

Second, we give the algorithm and analysis for Phase 2 (see Algorithm 3) to compute $q$ defined in Definition B.8.

**Lemma C.6** (Correctness of Phase 2). *The* ATTENTIONGRADIENTCACHEPHASE2 *(Algorithm 3) outputs a $n \times n$ matrix $q$ defined in Definition B.8.*

*Proof.* The algorithm first computes $h = A_3Y$. Then, it outputs $q = \mathrm{d}Oh^\top$ which is exactly the same as $q$ defined in Definition B.8. $\qquad\square$

**Lemma C.7** (I/O complexity of Phase 2). *The I/O complexity of* ATTENTIONGRADIENTCACHEP-HASE2 *(Algorithm 3) is* $O(\frac{n^2 d + n d^2}{\sqrt{M}})$.

*Proof.* In Phase 2 (Algorithm 3) the number of items in cache is at most $3B^2 \leq 4B^2 \leq M$. For each iteration in computing $h = A_3 Y$ and $q = \mathrm{d}O h^\top$, the algorithm reads $O(B^2)$ from memory into cache. This is the dominating factor of the I/O complexity of the algorithm. Thus, the I/O complexity of Phase 2 is $O(\frac{n^2 d}{B^3} B^2) + O(\frac{n d^2}{B^3} B^2) = O(\frac{n^2 d + n d^2}{B}) = O(\frac{n^2 d + n d^2}{\sqrt{M}})$. $\square$

---

**Algorithm 3** Attention gradient computation with cache phase 2. Compute $q$.

---

1: **procedure** ATTENTIONGRADIENTCACHEPHASE2($A_3, \mathrm{d}O \in \mathbb{R}^{n \times d}, f \in \mathbb{R}^{n \times n} Y \in \mathbb{R}^{d \times d}$,
   $M \in \mathbb{N}_+$)                                    ▷ Lemma C.6, Lemma C.7
2:    $B \leftarrow \lfloor \sqrt{M/4} \rfloor$
3:    /* Phase 2: Compute $q$ */
4:    **for** $1 \leq i \leq \lceil n/B \rceil$ **do**
5:       **for** $1 \leq j \leq \lceil d/B \rceil$ **do**
6:          Initialize $h^{(B)}[i, j] \leftarrow 0^{B \times B}$ in cache
7:          **for** $1 \leq k \leq \lceil d/B \rceil$ **do**
8:             Read $A_3^{(B)}[i, k]$ and $Y^{(B)}[k, j]$ into cache
9:             Compute $h^{(B)}[i, j] \leftarrow h^{(B)}[i, j] + A_3^{(B)}[i, k] Y^{(B)}[k, j]$ in cache
10:            Delete $A_3^{(B)}[i, k]$ and $Y^{(B)}[k, j]$ from cache
11:          **end for**
12:          Write $h^{(B)}[i, j]$ in to memory, and delete $h^{(B)}[i, j]$ from cache
13:       **end for**
14:    **end for**
15:    **for** $1 \leq i \leq \lceil n/B \rceil$ **do**
16:       **for** $1 \leq j \leq \lceil n/B \rceil$ **do**
17:          Initialize $q^{(B)}[i, j] \leftarrow 0^{B \times B}$ in cache
18:          **for** $1 \leq k \leq \lceil d/B \rceil$ **do**
19:             Read $\mathrm{d}O^{(B)}[i, k]$ and $(h^\top)^{(B)}[k, j]$ into cache
20:             Compute $q^{(B)}[i, j] \leftarrow q^{(B)}[i, j] + \mathrm{d}O^{(B)}[i, k] (h^\top)^{(B)}[k, j]$ in cache
21:            Delete $\mathrm{d}O^{(B)}[i, k]$ and $(h^\top)^{(B)}[k, j]$ from cache
22:          **end for**
23:          Write $q^{(B)}[i, j]$ in to memory, and delete $q^{(B)}[i, j]$ from cache
24:       **end for**
25:    **end for**
26:    **return** $q$                     ▷ $q \in \mathbb{R}^{n \times n}$, where $q$ is defined in Definiton B.8
27: **end procedure**

---

Then, we give the algorithm and analysis for Phase 3 (see Algorithm 4) to compute $p$ defined in Definition B.9.

**Lemma C.8** (Correctness of Phase 3). *The* ATTENTIONGRADIENTCACHEPHASE3 *(Algorithm 4) outputs a* $n \times n$ *matrix* $p$ *defined in Definition B.9.*

*Proof.* The algorithm first computes $v = (f \circ q) \cdot \mathbf{1}$. Then it outputs $p = f \circ q - \mathrm{diag}(v) f$. $\square$

**Lemma C.9** (I/O complexity of Phase 3). *The I/O complexity of* ATTENTIONGRADIENTCACHEP-HASE3 *(Algorithm 4) is* $O(\frac{n^2}{\sqrt{M}})$.

*Proof.* In Phase 3 (Algorithm 4) the number of items in cache is at most $3B^2 + B \leq 4B^2 \leq M$. For each iteration in computing $v = (f \circ q) \cdot \mathbf{1}$ and $p = f \circ q - \mathrm{diag}(v) f$. The algorithm reads $O(B^2)$ from memory into cache. This is the dominating factor of the I/O complexity of the algorithm. Thus, the I/O complexity of Phase 2 is $O(\frac{n^2}{B^3} B^2) = O(\frac{n^2}{B}) = O(\frac{n^2}{\sqrt{M}})$. $\square$

---

**Algorithm 4** Attention gradient computation with cache phase 3. Compute $p$.

---

1: **procedure** ATTENTIONGRADIENTCACHEPHASE3($q \in \mathbb{R}^{n \times n}, f \in \mathbb{R}^{n \times n}, M \in \mathbb{N}_+$) ▷ Lemma C.8, Lemma C.9
2:     $B \leftarrow \lfloor \sqrt{M/4} \rfloor$
3:     /* Phase 3: Compute $p$ */
4:     **for** $1 \leq i \leq \lceil n/B \rceil$ **do**
5:        Initialize $v^{(B)}[i] \leftarrow 0^B$ in cache
6:        **for** $1 \leq j \leq \lceil n/B \rceil$ **do**
7:           Read $f^{(B)}[i,j]$ and $q^{(B)}[i,j]$ into cache
8:           Compute $v^{(B)}[i] \leftarrow v^{(B)}[i] + (f^{(B)}[i,j] \circ q^{(B)}[i,j]) \cdot \mathbf{1}$      ▷ $v = (f \circ q) \cdot \mathbf{1}$
9:           Delete $f^{(B)}[i,j]$ and $q^{(B)}[i,j]$ from cache
10:        **end for**
11:        **for** $1 \leq j \leq \lceil n/B \rceil$ **do**
12:           Initialize $p^{(B)}[i,j] \leftarrow 0^{B \times B}$ in cache
13:           Read $f^{(B)}[i,j]$ and $q^{(B)}[i,j]$ into cache
14:           Compute $p^{(B)}[i,j] \leftarrow p^{(B)}[i,j] + f^{(B)}[i,j] \circ q^{(B)}[i,j] - \operatorname{diag}(v^{(B)}[i]) f^{(B)}[i,j]$
15:           Delete $f^{(B)}[i,j]$ and $q^{(B)}[i,j]$ from cache
16:           Write $p^{(B)}[i,j]$ in to memory, and delete $p^{(B)}[i,j]$ from cache
17:        **end for**
18:        Delete $v^{(B)}[i]$ from cache
19:     **end for**
20:     **return** $p$        ▷ $p \in \mathbb{R}^{n \times n}$, where $p$ is defined in Definiton B.9
21: **end procedure**

---

Lastly, we give the algorithm and analysis for Phase 4 (see Algorithm 5) to compute $\frac{\mathrm{d}L(X)}{\mathrm{d}X}$.

**Lemma C.10** (Correctness of Phase 4). *The* ATTENTIONGRADIENTCACHEPHASE4 *(Algorithm 5) outputs a $d \times d$ matrix $g = \frac{\mathrm{d}L(X)}{\mathrm{d}X}$ (Definition 3.2).*

*Proof.* The algorithm first computes $T = A_1^\top p$. Then it outputs $g = T A_2$. □

**Lemma C.11** (I/O complexity of Phase 4). *The I/O complexity of* ATTENTIONGRADIENTCACHEP-HASE4 *(Algorithm 5) is $O(\frac{n^2 d + n d^2}{\sqrt{M}})$.*

*Proof.* In Phase 4 (Algorithm 5) the number of items in cache is at most $3B^2 \leq 4B^2 \leq M$. For each iteration in computing $T = A_1^\top p$ and $g = T A_2$. The algorithm reads $O(B^2)$ from memory into cache. This is the dominating factor of the I/O complexity of the algorithm. Thus, the I/O complexity of Phase 2 is $O(\frac{n^2 d}{B^3} B^2) + O(\frac{n d^2}{B^3} B^2) = O(\frac{n^2 d + n d^2}{B}) = O(\frac{n^2 d + n d^2}{\sqrt{M}})$. □

## C.3 UPPER BOUND FOR ATTENTION BACKWARD IN SMALL CACHE $M = o(d^2)$

When cache size is not so big, i.e. $M = o(d^2)$, the attention backward is equivalent to matrix multiplication, thus having $O(\frac{n^2 d + n d^2}{\sqrt{M}})$ bound on the I/O complexity.

We show the upper bound theorem below for the overall algorithm (see Algorithm 6) to solve the attention backward in small cache case.

**Theorem C.12** (Small cache upper bound, formal version of Theorem 4.3). *Suppose $n$ is the input length, $d$ is the head dimension, and $M$ is the cache size. There is an algorithm (see Algorithm 6) outputs a $d \times d$ matrix $g = \frac{\mathrm{d}L(X)}{\mathrm{d}X}$ (Definition 3.2) with I/O complexity $O(\frac{n^2 d + n d^2}{\sqrt{M}})$, time complexity $\mathcal{T}_{\mathrm{mat}}(n, d, n) + \mathcal{T}_{\mathrm{mat}}(n, d, d)$, and space complexity $O(n^2 + d^2)$.*

*Proof.* **Time/space complexity.**

---

**Algorithm 5** Attention gradient computation with cache phase 4. Compute $\frac{\mathrm{d}L(X)}{\mathrm{d}X}$.

---

1: **procedure** ATTENTIONGRADIENTCACHEPHASE4($A_1, A_2 \in \mathbb{R}^{n \times d}, p \in \mathbb{R}^{n \times n}, M \in \mathbb{N}_+$)  ▷
Lemma C.10, Lemma C.11
2:     $B \leftarrow \lfloor \sqrt{M/4} \rfloor$
3:     /* Phase 4: Compute $\frac{\mathrm{d}L(X)}{\mathrm{d}X}$ */
4:     **for** $1 \le i \le \lceil d/B \rceil$ **do**
5:         **for** $1 \le j \le \lceil n/B \rceil$ **do**
6:             Initialize $T^{(B)}[i,j] \leftarrow 0^{B \times B}$ in cache
7:             **for** $1 \le k \le \lceil n/B \rceil$ **do**
8:                 Read $(A_1^\top)^{(B)}[i,k]$ and $p^{(B)}[k,j]$ into cache
9:                 Compute $T^{(B)}[i,j] \leftarrow T^{(B)}[i,j] + (A_1^\top)^{(B)}[i,k]p^{(B)}[k,j]$ in cache ▷ $T = A_1^\top p$
10:                Delete $(A_1^\top)^{(B)}[i,k]$ and $p^{(B)}[k,j]$ from cache
11:             **end for**
12:             Write $T^{(B)}[i,j]$ in to memory, and delete $T^{(B)}[i,j]$ from cache
13:         **end for**
14:     **end for**
15:     **for** $1 \le i \le \lceil d/B \rceil$ **do**
16:         **for** $1 \le j \le \lceil d/B \rceil$ **do**
17:             Initialize $g^{(B)}[i,j] \leftarrow 0^{B \times B}$ in cache
18:             **for** $1 \le k \le \lceil n/B \rceil$ **do**
19:                 Read $T^{(B)}[i,k]$ and $A_2^{(B)}[k,j]$ into cache
20:                 Compute $g^{(B)}[i,j] \leftarrow g^{(B)}[i,j] + T^{(B)}[i,k]A_2^{(B)}[k,j]$ in cache     ▷ $g = TA_2$
21:                Delete $T^{(B)}[i,k]$ and $A_2^{(B)}[k,j]$ from cache
22:             **end for**
23:             Write $g^{(B)}[i,j]$ in to memory, and delete $g^{(B)}[i,j]$ from cache
24:         **end for**
25:     **end for**
26:     **return** $g$            ▷ $g = \frac{\mathrm{d}L(X)}{\mathrm{d}X} \in \mathbb{R}^{d \times d}$, see Definition 3.2
27: **end procedure**

---

First, we notice that Algorithm 6 calculates the same gradients as the Algorithm 1 except that the former utilize cache to speed up the computation and specify the standard matrix multiplication computations in cache. Thus, the overall time complexity $\mathcal{T}_{\mathrm{mat}}(n, d, n) + \mathcal{T}_{\mathrm{mat}}(n, d, d)$, and space complexity $O(n^2 + d^2)$ should be the same as Lemma C.3.

**I/O complexity.**

From Lemma C.5, C.7, C.9, and C.11, we know the overall I/O complexity is $O(\frac{n^2 d + nd^2}{\sqrt{M}}) + O(\frac{n^2}{\sqrt{M}}) = O(\frac{n^2 d + nd^2}{\sqrt{M}})$.

**Correctness.**

From Lemma C.4, C.6, C.8, and C.10, the algorithm computes the correct $\frac{\mathrm{d}L(X)}{\mathrm{d}X}$.     □

---

**Algorithm 6** Attention gradient computation with small cache.

---

1: **procedure** ATTENTIONGRADIENTCACHE($A_1, A_2, A_3, \mathrm{d}O \in \mathbb{R}^{n \times d}, X, Y \in \mathbb{R}^{d \times d}, M \in \mathbb{N}_+$)
    ▷ Theorem C.12
2:     $f \leftarrow$ ATTENTIONGRADIENTCACHEPHASE1($A_1, A_2, X, M$)         ▷ see Algorithm 2
3:     $q \leftarrow$ ATTENTIONGRADIENTCACHEPHASE2($A_3, \mathrm{d}O, f, Y, M$)     ▷ see Algorithm 3
4:     $p \leftarrow$ ATTENTIONGRADIENTCACHEPHASE3($q, f, M$)         ▷ see Algorithm 4
5:     $g \leftarrow$ ATTENTIONGRADIENTCACHEPHASE4($A_1, A_2, p, M$)       ▷ see Algorithm 5
6:     **return** $g$           ▷ $g = \frac{\mathrm{d}L(X)}{\mathrm{d}X} \in \mathbb{R}^{d \times d}$, see Definition 3.2
7: **end procedure**

---

# D   I/O COMPLEXITY UPPER BOUND FOR LARGE CACHE

In this section, we establish the upper bound (Theorem D.5) for the I/O complexity in the case where the cache size is large, specifically when $M = \Omega(d^2)$. Section D.1 presents algorithms and analyses for attention gradient computation in the large cache setting. Section D.2 provides the upper bound theorem for the large cache case.

Since our goal is to compute the backward pass of the attention mechanism, and the forward pass has already been performed, it is natural to assume that we have access to the softmax normalizing vector $l := A \cdot \mathbf{1} \in \mathbb{R}^n$ (Definition B.4) and the final attention forward output $O = \mathrm{diag}(l)^{-1} AV \in \mathbb{R}^{n \times d}$ (Definition B.7) where $A = \exp(A_1 X A_2^\top)$ (Definition B.3).

By utilizing these precomputed quantities from the forward pass, we can efficiently proceed with the backward computation while optimizing the I/O operations required.

## D.1   ALGORITHMS FOR ATTENTION BACKWARD IN LARGE CACHE

We first give Algorithm 7 and its analysis in large cache case for computing intermediate variables $S, h$.

---

**Algorithm 7** Attention gradient computation large cache phase 1. Compute $S, h$.

---

1: **procedure** ATTENTIONGRADIENTLARGECACHEPHASE1($A_1, A_3 \in \mathbb{R}^{n \times d}$, $X, Y \in \mathbb{R}^{d \times d}$, $M \in \mathbb{N}_+$)                                    $\triangleright$ Lemma D.1, Lemma D.2
2:     $B_r \leftarrow \min\{\lceil \frac{M}{4d} \rceil, d\}$ and $B_c \leftarrow \lceil \frac{M}{4d} \rceil$
3:     Vertically divide $A_1$ into $T_r = \lceil \frac{n}{B_r} \rceil$ blocks $A_{1,1}, \ldots, A_{1,T_r}$ of size $B_r \times d$ each, and horizontally divide $X$ into $T_c = \lceil \frac{d}{B_c} \rceil$ blocks $X_{*,1}, \ldots, X_{*,T_c}$ of size $d \times B_c$ each
4:     Vertically divide $A_3$ into $T_r = \lceil \frac{n}{B_r} \rceil$ blocks $A_{3,1}, \ldots, A_{3,T_r}$ of size $B_r \times d$ each, and horizontally divide $Y$ into $T_c = \lceil \frac{d}{B_c} \rceil$ blocks $Y_{*,1}, \ldots, Y_{*,T_c}$ of size $d \times B_c$ each
5:                    $\triangleright$ Here $A_{1,i}, A_{3,i} \in \mathbb{R}^{B_r \times d}$ means the $i$-th row block of $A_1, A_3$ for $i \in [T_r]$, and $X_{*,j}, Y_{*,j} \in \mathbb{R}^{d \times B_c}$ means $j$-th column block of $X, Y$ for $j \in [T_c]$
6:     **for** $1 \leq i \leq T_r$ **do**
7:         Read $A_{1,i}, A_{3,i} \in \mathbb{R}^{B_r \times d}$ into cache
8:         **for** $1 \leq j \leq T_c$ **do**
9:             Read $X_{*,j} \in \mathbb{R}^{d \times B_c}$ into cache, and initialize $S_{i,j} \leftarrow 0^{B_r \times B_c}$ in cache
10:            Compute $S_{i,j} \leftarrow S_{i,j} + A_{1,i} X_{*,j}$ in cache                      $\triangleright S = A_1 X$
11:            Write $S_{i,j}$ to memory, and delete $S_{i,j}, X_{*,j}$ from cache
12:            Read $Y_{*,j} \in \mathbb{R}^{d \times B_c}$ into cache, and initialize $h_{i,j} \leftarrow 0^{B_r \times B_c}$ in cache
13:            Compute $h_{i,j} \leftarrow h_{i,j} + A_{3,i} Y_{*,j}$ in cache                      $\triangleright h = A_3 Y$
14:            Write $h_{i,j}$ to memory, and delete $h_{i,j}, Y_{*,j}$ from cache
15:        **end for**
16:        Delete $A_{1,i}, A_{3,i}$ from cache
17:    **end for**
18:    **return** $S, h$                                    $\triangleright S, h \in \mathbb{R}^{n \times d}$
19: **end procedure**

---

**Lemma D.1** (Correctness of Phase 1). *The* ATTENTIONGRADIENTLARGECACHEPHASE1 *(Algorithm 7) outputs two $n \times d$ matrices $S = A_1 X$ (Definition 3.1) and $h = A_3 Y$ (Definition B.6).*

*Proof.* The algorithm first divide $A_1, A_3, X, Y$ into row/column blocks of size $B_r \times d$ or $d \times B_c$. Then it reads the row/column block matrices to compute the corresponding small blocks of $S, h$ by standard matrix multiplication. Thus, it computes the exact value for $S, h$.                    $\square$

**Lemma D.2** (I/O complexity of Phase 1). *Suppose the cache size satisfy $nd \geq M \geq d$. The I/O complexity of* ATTENTIONGRADIENTLARGECACHEPHASE1 *(Algorithm 7) is $O(\frac{n^2 d^2}{M} + \frac{n d^3}{M})$.*

*Proof.* **Why such conditions for $B_r, B_c$.**

The cache size has three constraints, because we need matrices $A_{1,i}, A_{3,i} \in \mathbb{R}^{B_r \times d}$, $X_{*,j}, Y_{*,j} \in \mathbb{R}^{d \times B_c}$, and $S_{i,j}, h_{i,j} \in \mathbb{R}^{B_r \times B_c}$ to fit into cache. Thus, we have

$$B_r d = O(M)$$
$$B_c d = O(M)$$
$$B_r B_c = O(M)$$

Then, we need

$$B_r = O(M/d)$$
$$B_c = O(M/d)$$

By setting $B_c = \Theta(M/d)$, we have

$$B_r = \Theta(\min\{M/d, M/B_c\})$$
$$= \Theta(\min\{M/d, d\})$$

**I/O complexity.** We know $B_r \leftarrow \min\{\lceil \frac{M}{4d} \rceil, d\}$ and $B_c \leftarrow \lceil \frac{M}{4d} \rceil$, also $T_r = \lceil \frac{n}{B_r} \rceil$ and $T_c = \lceil \frac{d}{B_r} \rceil$. Substituting $B_r$ into $T_r$, we get $T_r = O(\frac{nd}{M})$. Observe that $T_r B_r = O(n)$ and $T_c B_c = O(d)$.

The I/O complexity can be computed by:

$$T_r(B_r d + T_c(dB_c)) = O(nd) + T_r d^2$$
$$= O(nd) + O(\frac{nd}{M} d^2)$$
$$= O(nd + \frac{nd^3}{M})$$

where the first step follows from $T_r B_r = O(n)$ and $T_c B_c = O(d)$, the second step follows from $T_r = O(\frac{nd}{M})$, and the last step follows from simple algebra.

Because $M \leq nd$, we have

$$O(nd + \frac{nd^3}{M}) = O(\frac{ndM}{M} + \frac{nd^3}{M})$$
$$= O(\frac{n^2 d^2}{M} + \frac{nd^3}{M})$$

Thus, the total I/O complexity is $O(\frac{n^2 d^2}{M} + \frac{nd^3}{M})$  $\square$

We then give Algorithm 8 along with its analysis for computing the gradient $g$.

**Lemma D.3** (Correctness of Phase 2). *The* ATTENTIONGRADIENTLARGECACHEPHASE2 *(Algorithm 8) outputs a $d \times d$ matrix $g$ (Definition 3.2).*

*Proof.* The algorithm first vertically divides the matrices $S, A_2, l, O, \mathrm{d}O, h$, and $A_1$ into row blocks of size $B_r \times d$ or $B_c \times d$. Following the computational graph (Fig. 2) and the no-cache algorithm (Algorithm 1), we compute the gradient $g$ exactly. It is important to note that, in algorithm design, we need to avoid reading the attention matrix $f \in \mathbb{R}^{n \times n}$ directly—even though it has been computed during the forward pass—or any matrices of size $B_r \times n$ or $B_c \times n$. Doing so would result in an $O(n^2)$ I/O complexity, which cannot be improved through caching.  $\square$

**Lemma D.4** (I/O complexity of Phase 2). *Suppose the cache size satisfy $nd \geq M \geq d^2$. The I/O complexity of* ATTENTIONGRADIENTLARGECACHEPHASE2 *(Algorithm 8) is $O(\frac{n^2 d^2}{M} + \frac{nd^3}{M})$.*

*Proof.* The reason for conditions of $B_r, B_c$ is the same as the proof of Lemma D.2. However, it is important to note that updating the gradient $g$ in the cache requires assuming a cache size of $M \geq d^2$. This is necessary because we fuse the key and query weight matrices into a single matrix $X \in \mathbb{R}^{d \times d}$. The update to the corresponding gradient $g$ in the cache is driven by the outer product representation of the matrix, as shown in Line 21 of Algorithm 8.

---

**Algorithm 8** Attention gradient computation large cache phase 2. Compute $g$.

---

1: **procedure** ATTENTIONGRADIENTLARGECACHEPHASE2($A_1, A_2, S, h, O, \mathrm{d}O \in \mathbb{R}^{n \times d}, l \in \mathbb{R}^n, M \in \mathbb{N}_+$)          $\triangleright$ Lemma D.3, Lemma D.4

2:      $B_r \leftarrow \min\{\lceil \frac{M}{4d} \rceil, d\}$ and $B_c \leftarrow \lceil \frac{M}{4d} \rceil$

3:      Vertically divide $S$ into $T_r = \lceil \frac{n}{B_r} \rceil$ blocks $S_1, \ldots, S_{T_r}$ of size $B_r \times d$ each, vertically divide $A_2$ into $T_c = \lceil \frac{n}{B_c} \rceil$ blocks $A_{2,1}, \ldots, A_{2,T_c}$ of size $B_c \times d$ each, and vertically divide $l$ into $T_r = \lceil \frac{n}{B_r} \rceil$ blocks $l_1, \ldots, l_{T_r}$ of size $B_r$ each

4:      Vertically divide $O$ into $T_r = \lceil \frac{n}{B_r} \rceil$ blocks $O_1, \ldots, O_{T_r}$ of size $B_r \times d$ each, vertically divide $\mathrm{d}O$ into $T_r = \lceil \frac{n}{B_r} \rceil$ blocks $\mathrm{d}O_1, \ldots, \mathrm{d}O_{T_r}$ of size $B_r \times d$ each, vertically divide $h$ into $T_c = \lceil \frac{n}{B_c} \rceil$ blocks $h_1, \ldots, h_{T_c}$ of size $B_c \times d$ each, and vertically divide $A_1$ into $T_r = \lceil \frac{n}{B_r} \rceil$ blocks $A_{1,1}, \ldots, A_{1,T_r}$ of size $B_r \times d$ each

5:      Initialize $g \leftarrow 0^{d \times d}$ in cache

6:      **for** $1 \le i \le T_r$ **do**

7:          Read $S_i, O_i, \mathrm{d}O_i, A_{1,i} \in \mathbb{R}^{B_r \times d}$ and $l_i \in \mathbb{R}^{B_r}$ into cache

8:          Initialize $v_i \leftarrow 0^{B_r}$ and compute $v_i \leftarrow v_i + (\mathrm{d}O_i \circ O_i) \cdot \mathbf{1}$ in cache $\triangleright v = (\mathrm{d}O \circ O) \cdot \mathbf{1}$

9:          Delete $O_i$ from cache

10:          **for** $1 \le j \le T_c$ **do**

11:              Read $h_j \in \mathbb{R}^{B_c \times d}$ and initialize $q_{i,j} \leftarrow 0^{B_r \times B_c}$ in cache

12:              Compute $q_{i,j} \leftarrow \mathrm{d}O_i h_j^\top$ in cache          $\triangleright q = \mathrm{d}O h^\top$

13:              Read $A_{2,j} \in \mathbb{R}^{B_c \times d}$ into cache, and initialize $A_{i,j} \leftarrow 0^{B_r \times B_c}$ in cache

14:              Compute $A_{i,j} \leftarrow A_{i,j} + S_i A_{2,j}^\top$ in cache          $\triangleright A = S A_2^\top$

15:              Compute $A_{i,j} \leftarrow \exp(A_{i,j})$ in cache, and initialize $f_{i,j} \leftarrow 0^{B_r \times B_c}$ in cache

16:              Compute $f_{i,j} \leftarrow f_{i,j} + \mathrm{diag}(l_i)^{-1} A_{i,j}$ in cache      $\triangleright f = \mathrm{diag}(l) A$

17:              Delete $A_{i,j}$ from cache, and initialize $p_{i,j} \leftarrow 0^{B_r \times B_c}$ in cache

18:              Compute $p_{i,j} \leftarrow p_{i,j} + f_{i,j} \circ q_{i,j} - \mathrm{diag}(v_i) f_{i,j}$ in cache    $\triangleright p = f \circ q - \mathrm{diag}(v) f$

19:              Delete $f_{i,j}, q_{i,j}$ in cache, and initialize $T_{*,j} \leftarrow 0^{d \times B_c}$ in cache

20:              Compute $T_{*,j} \leftarrow T_{*,j} + A_{1,i}^\top p_{i,j}$ in cache          $\triangleright T = A_1^\top p$

21:              Compute $g \leftarrow g + T_{*,j} A_{2,j}$ in cache          $\triangleright g = T A_2$

22:              Delete $T_{*,j}, A_{2,j}$ from cache

23:          **end for**

24:          Delete $S_i, A_{1,i}, \mathrm{d}O_i, l_i, v_i$ from cache

25:      **end for**

26:      Write $g$ into memory

27:      **return** $g$          $\triangleright g = \frac{\mathrm{d}L(X)}{\mathrm{d}X} \in \mathbb{R}^{d \times d}$, see Definition 3.2

28: **end procedure**

---

Next we show the I/O complexity. Since $B_r \leftarrow \min\{\lceil \frac{M}{4d} \rceil, d\}$ and $B_c \leftarrow \lceil \frac{M}{4d} \rceil$, also $T_r = \lceil \frac{n}{B_r} \rceil$ and $T_c = \lceil \frac{n}{B_r} \rceil$, we get $T_r = O(\frac{nd}{M})$. Also, we observe that $T_r B_r = O(n)$ and $T_c B_c = O(n)$.

The I/O complexity can be computed by:

$$T_r(B_r d + T_c B_c d) + d^2 = O(nd) + T_r n d + d^2$$
$$= O(T_r n d) + d^2$$
$$= O(\frac{n^2 d^2}{M}) + d^2$$

where the first step follows from $T_r B_r = O(n)$ and $T_c B_c = O(n)$, the second step follows from $T_r \ge 1$, and the last step follows from $T_r = O(\frac{nd}{M})$.

Then, because $M \le nd$, we can show

$$O(d^2 + \frac{n^2 d^2}{M}) = O(\frac{d^2 M}{M} + \frac{n^2 d^2}{M})$$
$$= O(\frac{nd^3}{M} + \frac{n^2 d^2}{M})$$

Thus, the total I/O complexity is $O(\frac{n^2 d^2}{M} + \frac{nd^3}{M})$ □

## D.2 Upper Bound for Attention Backward in Large Cache $M = \Omega(d^2)$

In the large cache scenario, while it is feasible to precompute and store the $n \times n$ attention matrix, reading it will result in an unavoidable $O(n^2)$ I/O complexity. Inspired by FlashAttention (Dao et al., 2022; Dao, 2023; Shah et al., 2024), we present the following theorem, which provides an upper bound $O(\frac{n^2 d^2 + nd^3}{M})$ on the I/O complexity of the attention gradient algorithm in the large cache (Algorithm 9).

**Theorem D.5** (Large cache upper bound, formal version of Theorem 4.1). *Suppose $n$ is the input length, $d$ is the head dimension, and $nd \geq M \geq d^2$ is the cache size. There is an algorithm (see Algorithm 9) outputs a $d \times d$ matrix $g = \frac{\mathrm{d}L(X)}{\mathrm{d}X}$ (Definition 3.2) with I/O complexity $O(\frac{n^2 d^2 + nd^3}{M})$.*

*Proof.* **Correctness.** Combining Lemma D.1 and D.3, we finish the proof.

**I/O complexity.** Combining Lemma D.2 and D.4, we finish the proof. □

---

**Algorithm 9** Attention gradient computation with large cache.

1: **procedure** ATTENTIONGRADIENTLARGECACHE($A_1, A_2, A_3, O, \mathrm{d}O \in \mathbb{R}^{n \times d}, X, Y \in \mathbb{R}^{d \times d},$
   $l \in \mathbb{R}^n, M \in \mathbb{N}_+$) ▷ Theorem D.5
2:  $S, h \leftarrow$ ATTENTIONGRADIENTLARGECACHEPHASE1($A_1, A_3, X, Y, M$) ▷ see
   Algorithm 7
3:  $g \leftarrow$ ATTENTIONGRADIENTLARGECACHEPHASE4($A_1, A_2, h, S, O, \mathrm{d}O, l, M$) ▷ see
   Algorithm 8
4:  **return** $g$ ▷ $g = \frac{\mathrm{d}L(X)}{\mathrm{d}X} \in \mathbb{R}^{d \times d}$, see Definition 3.2
5: **end procedure**

---

## E Lower Bound for Attention Backward Computation

In this section, we prove the lower bound of the attention gradient computation. In Section E.1, we state some definition in graph theory that will be used to establish the framework of (Hong & Kung, 1981) that will be used to analyze the I/O complexity. In Section E.2, we state some tools from previous works from I/O compleixty of standard matrix multiplication and attention forward computation. In Section E.3, we will establish our lower bounds of I/O complexity for attention backward passes in both large cache case and small cache case.

### E.1 Basic Definition in Graph Theory

Hong & Kung (1981) introduces a method for analyzing I/O complexity using the concept of an $M$-partition on a graph. Before we define it, we first provide some definitions from graph theory.

**Definition E.1** (Dominator set). *Let $G = (V, E)$ be a directed acyclic graph and $S \subseteq V$. We define a set $D \subseteq V$ as a dominator set of $S$ if, for every path in $G$ from a input node to any node in $S$, there exists at least one node in $D$ on that path.*

**Definition E.2** (Minimum set). *Let $G = (V, E)$ be a directed acyclic graph and $S \subseteq V$. We say that a set $M \subseteq S$ is a minimum set of $S$ if $M$ contains all nodes in $S$ that have no children in $S$.*

**Definition E.3** (Vertex subset dependence). *Let $G = (V, E)$ be a directed acyclic graph. Let $V_1, V_2 \subseteq V$ be two disjoint subsets. We say that $V_2$ depends on $V_1$ if there is a directed edge from a node in $V_1$ to a node in $V_2$.*

**Definition E.4** (Cyclic dependence). *Let $G = (V, E)$ be a directed acyclic graph. Let $V_1, \ldots, V_h \subseteq V$ be $h$ disjoint subsets of $V$. We say that there is a cyclic dependence among $\{V_1, \ldots, V_h\}$ if there exists a permutation $(i_1, \ldots, i_h)$ of $[h]$ such that $V_{i_1}$ depends on $V_{i_h}$, and for every $j \in \{2, \ldots, h\}$, $V_{i_j}$ depends on $V_{i_{j-1}}$.*

Now, we are ready to define $M$-partitons. In fact, the minimum number of sets in any $M$-partition provides a lower bound on the I/O complexity.

**Definition E.5** ($M$-partition (Hong & Kung, 1981)). *Let $G = (V, E)$ be a directed acyclic graph. Let $V_1, \ldots, V_h \subseteq V$ be $h$ disjoint subsets of $V$. We say that $\{V_1, \ldots, V_h\}$ is a $M$-partition of $G$ if the following conditions are satisfied*

- *$\{V_1, \ldots, V_h\}$ is a partition of $V$, i.e., $V_1, \ldots, V_h$ are disjoint and $V = \bigcup_{i=1}^{h} V_i$.*

- *For each $V_i$, there exists a dominator set $D_i$ of $V_i$ such that $D_i$ has at most $M$ nodes.*

- *For each $V_i$, there exists a minimum set $M_i$ of $V_i$ such that $M_i$ has at most $M$ nodes.*

- *There is no cyclic dependence among $\{V_1, \ldots, V_h\}$.*

*We use $P(G, M)$ to denote the minimum number of sets in any $M$-partition of $G$.*

### E.2 Previous Tools for I/O Complexity

Now, we are ready to introduce some tools for I/O Complexity from Hong & Kung (1981) by using an $M$-partition on a graph.

**Lemma E.6** (Lemma 3.1 of Hong & Kung (1981)). *For any directed acyclic graph $G$ and any positive integer $M$, we have*

$$Q(G, M) \geq M \cdot (P(G, 2M) - 1).$$

*We omit $G$ when it is clear in the context.*

We state two useful lemmas from previous works as follows.

**Lemma E.7** (Lemma 3.3 of Saha & Ye (2024)). *Suppose that $M = \Omega(d^2)$ and $A \in \mathbb{R}^{n_1 \times d}, B \in \mathbb{R}^{d \times n_2}$. Let $\mathcal{P}$ be an $M$-partition of the computational graph of any algorithm that computes $AB$ using standard matrix multiplication. Then for each $V' \in \mathcal{P}$, $V'$ contains at most $O(\frac{M^2}{d})$ product nodes $A_{i,k} B_{k,j}$, sum nodes $(AB)_{i,j}$, and all intermediate nodes in the summation trees.*

In Saha & Ye (2024), the matrices $A$ and $B$ in the above lemma are of sizes $n \times d$ and $d \times n$, respectively. We note that with slight modifications to the proofs, the result also holds when $A$ and $B$ have different sizes, specifically $n_1 \times d$ and $d \times n_2$.

The next lemma gives the lower bound of I/O compleixty of standard matrix multiplication.

**Lemma E.8** (Corollary 6.2 of Hong & Kung (1981)). *Let $A \in \mathbb{R}^{n_1 \times d}, B \in \mathbb{R}^{d \times n_2}$. The standard matrix multiplication algorithm computing $AB$ has I/O complexity $Q(M) = \Omega(\frac{n_1 d n_2}{\sqrt{M}})$.*

### E.3 Proof of Our Lower Bound

We establish the lower bounds of I/O complexity of attention gradient computation in both large cache case and small cache case. We first give the lower bound in the large cache case, i.e., the cache size $M = \Omega(d^2)$.

**Theorem E.9** (Large cache lower bound, formal version of Theorem 4.2). *Suppose $n$ is the input length and $d$ is the head dimension. Suppose the cache size $M = \Omega(d^2)$. Then the I/O complexity of attention gradient computation using standard matrix multiplication is $\Omega(\frac{n^2 d^2 + n d^3}{M})$.*

*Proof.* Any algorithm that computes the attention gradient needs to compute the matrix product $A_1 X A_2^\top$ using standard matrix multiplication. Note that we compute $A_1 X A_2^\top$ using standard matrix multiplication, so we either first compute $A_1 X$ and then compute $(A_1 X) A_2^\top$, or first compute $X A_2^\top$ and then compute $A_1 (X A_2^\top)$. In either case, we perform two matrix multiplications: one between an $n \times d$ matrix and a $d \times d$ matrix, and another between an $n \times d$ matrix and a $d \times n$ matrix. Without loss of generality, we assume the first case where we first compute $A_1 X$.

Recall that the level-1 nodes are the product nodes $(A_1)_{i,k} X_{k,j}$, the sum nodes $(A_1 X)_{i,j}$, and all intermediate nodes in the summation trees. For every $V'$ in an $M$-partition $\mathcal{P}$, by Lemma E.7, there

are at most $O(\frac{M^2}{d})$ level-1 nodes in $V'$. Since the number of sum nodes $(A_1X)_{i,j}$ is $nd^2$, the number of parts in the $M$-partition $\mathcal{P}$ is at least $\Omega(\frac{nd^3}{M^2})$. By Lemma E.6, the I/O complexity for computing $A_1X$ is $\Omega(\frac{n^2d}{M})$.

Similarly, we recall that level-2 nodes are the product nodes $(A_1X)_{i,k}(A_2^\top)_{k,j}$, the sum nodes $((A_1X)A_2^\top)_{i,j}$, and all intermediate nodes in the summation trees. For every $V'$ in an $M$-partition $\mathcal{P}$, by Lemma E.7, there are at most $O(\frac{M^2}{d})$ level-2 nodes in $V'$. Since the number of sum nodes $((A_1X)A_2^\top)_{i,j}$ is $n^2d$, the number of parts in the $M$-partition $\mathcal{P}$ is at least $\Omega(\frac{n^2d^2}{M^2})$. By Lemma E.6, the I/O complexity for computing $(A_1X)A_2^\top$ is $\Omega(\frac{n^2d^2}{M})$.

Therefore, the I/O complexity of attention gradient computation is at least $\Omega(\frac{nd^3+n^2d^2}{M})$. $\qquad\square$

Next, we give the lower bound in the small cache case, i.e., the cache size $M = o(d^2)$.

**Theorem E.10** (Small cache lower bound, formal version of Theorem 4.4). *Suppose $n$ is the input length and $d$ is the head dimension. Suppose the cache size $M = o(d^2)$. Then the I/O complexity of attention gradient computation using standard matrix multiplication is $\Omega(\frac{n^2d+nd^2}{\sqrt{M}})$.*

*Proof.* We show that when $M = o(d^2)$, the attention gradient computation can be reduced to computing the matrix product $A_1XA_2^\top$. Note that we compute $A_1XA_2^\top$ using standard matrix multiplication, so we either compute $A_1X$ first and then compute $(A_1X)A_2^\top$, or we first compute $XA_2^\top$ and then $A_1(XA_2^\top)$. However, both cases require performing one matrix multiplication between an $n \times d$ matrix and a $d \times d$ matrix, and one matrix multiplication between an $n \times d$ matrix and a $d \times n$ matrix. Hence, without loss of generality, we assume that $A_1X$ is computed first. By Lemma E.8, the I/O complexity of computing $A_1X$ is $\Omega(\frac{nd^2}{\sqrt{M}})$, and the I/O complexity of computing $(A_1X)A_2^\top$ is $\Omega(\frac{n^2d}{\sqrt{M}})$. Hence, the total I/O complexity of computing $A_1XA_2^\top$ is $\Omega(\frac{n^2d+nd^2}{\sqrt{M}})$.

Suppose that there is an algorithm $\mathcal{A}$ for attention gradient computation which has I/O complexity $o(\frac{n^2d+nd^2}{\sqrt{M}})$. We construct an algorithm $\mathcal{B}$ that computes the matrix product $A_1XA_2^\top$ with I/O complexity $o(\frac{n^2d+nd^2}{\sqrt{M}})$. Since $M < o(d^2)$, we have $\frac{n^2d+nd^2}{\sqrt{M}} > \omega(n^2 + nd) > \omega(n^2)$, so algorithm $\mathcal{A}$ is able to transfer the all entries of matrix product $(A_1X)A_2^\top$ from cache to memory. In the language of the red-blue pebble game, algorithm $\mathcal{B}$ works as follows: whenever algorithm $\mathcal{A}$ delete a blue pebble from a node in $(A_1X)A_2^\top$, do not delete it; whenever algorithm $\mathcal{A}$ place a red pebble on a node in $(A_1X)A_2^\top$, also place a blue pebble on it. Since the I/O complexity of algorithm $\mathcal{A}$ is $o(\frac{n^2d+nd^2}{\sqrt{M}})$ and we need an additional $n^2$ I/O operations to transfer the entries of the matrix product $(A_1X)A_2^\top$ from cache to memory. Since $n^2 < o(\frac{n^2d}{\sqrt{M}})$, the overall I/O complexity of $\mathcal{B}$ is still $o(\frac{n^2d+nd^2}{\sqrt{M}})$. However, this contradicts the fact that the I/O complexity of computing $A_1XA_2^\top$ is $\Omega(\frac{n^2d+nd^2}{\sqrt{M}})$. Therefore, the I/O complexity of attention gradient computation using standard matrix multiplication is $\Omega(\frac{n^2d+nd^2}{\sqrt{M}})$. $\qquad\square$

# F   SPARSE ATTENTION COMPUTATION

In this section, we provide the lower bounds of sparse attention computation for both forward and backward passes. In Section F.1, we state previous tools of sparse matrix multiplication. In Section F.2, we provide the proofs of the lower bounds of sparse attention.

## F.1   PREVIOUS TOOLS FOR I/O COMPLEXITY OF SPARSE MATRIX MULTIPLICATION

We assume that sparse matrices are stored by listing only their non-zero entries along with their coordinates. Sparse semi-ring matrix multiplication restricts operations to addition and multiplication of these entries, which means that each output entry $(AB)_{i,j}$ can only be computed as the sum of products given by $\sum_k A_{i,k}B_{k,j}$.

**Lemma F.1** (Theorem 2 of (Pagh & Stöckel, 2014)). *Let $A \in \mathbb{R}^{n_1 \times d}$ and $B \in \mathbb{R}^{d \times n_2}$ be two matrices such that $R_1 := \mathrm{nnz}(A) + \mathrm{nnz}(B)$ and $R_2 := \mathrm{nnz}(AB)$. The sparse semi-ring matrix multiplication that computes $AB$ has I/O complexity $\Omega(\min\{\frac{R_1^2}{M}, \frac{R_1 \sqrt{R_2}}{\sqrt{M}}\})$.*

Note that in this statement, the I/O complexity also separates into the large cache case and the small cache case, but the dividing point may not be $d^2$. It depends on whether all the necessary values for computing each output entry can be stored in the cache during the computation.

### F.2 OUR LOWER BOUNDS FOR SPARSE ATTENTION COMPUTATION

We first prove a useful lemma which state the lower bound of I/O complexity of computing the attention matrix.

**Lemma F.2.** *Let $A_1 \in \mathbb{R}^{n \times d}, X \in \mathbb{R}^{d \times d}, A_2 \in \mathbb{R}^{d \times n}$ be three matrices. Let $Z_A := \min\{\mathrm{nnz}(A_1), \mathrm{nnz}(A_2)\}, Z_X := \mathrm{nnz}(X), Z_{AX} = \min\{\mathrm{nnz}(A_1 X), \mathrm{nnz}(X A_2^\top)\}, Z_{AXA} := \mathrm{nnz}(A_1 X A_2^\top)$. Then the sparse semi-ring matrix multiplication that computes $A_1 X A_2^\top$ has I/O complexity $\Omega(\min\{\frac{Z_A^2 + Z_A Z_X}{M}, \frac{Z_A \sqrt{Z_{AXA}} + \sqrt{Z_A Z_X Z_{AX}}}{\sqrt{M}}\})$.*

*Proof.* We first consider the case where all the necessary values for computing each output entry can be stored in the cache during the computation. Suppose that $A_1 X$ is computed first, by Lemma F.1, computing $A_1 X$ has I/O compleixty

$$\Omega(\frac{(\mathrm{nnz}(A_1) + \mathrm{nnz}(X))^2}{M}) = \Omega(\frac{\mathrm{nnz}(A_1)^2 + 2 \mathrm{nnz}(A_1) \mathrm{nnz}(X) + \mathrm{nnz}(X)^2}{M})$$

$$\geq \Omega(\frac{Z_A^2 + 2 Z_A Z_X + Z_X^2}{M})$$

$$\geq \Omega(\frac{Z_A^2 + 2 Z_A Z_X}{M})$$

where the first step follows by the basic algebra, the second step uses the definition of $Z_A, Z_X$, and the last step follows from the basic algebra. Then we compute the product $(A_1 X) A_2^\top$, by Lemma F.1, computing $A_1 X$ has I/O compleixty

$$\Omega(\frac{(\mathrm{nnz}(A_1 X) + \mathrm{nnz}(A_2))^2}{M}) = \Omega(\frac{\mathrm{nnz}(A_1 X)^2 + 2 \mathrm{nnz}(A_1 X) \mathrm{nnz}(A_2) + \mathrm{nnz}(A_2)^2}{M})$$

$$\geq \Omega(\frac{\mathrm{nnz}(A_2)^2}{M})$$

$$= \Omega(\frac{Z_A^2}{M})$$

where the first and second steps follow by the basic algebra, and the last step uses the definition of $Z_A$. Therefore, computing $A_1 X A_2^\top$ in this way has I/O complexity $\Omega(\frac{2 Z_1^2 + 2 Z_1 Z_2}{M}) = \Omega(\frac{Z_1^2 + Z_1 Z_2}{M})$. Similary, suppose that $X A_2^\top$ is computed first. Then we can also get the I/O complexity $\Omega(\frac{Z_1^2 + Z_1 Z_2}{M})$.

Next, we consider the case where some elementary products of matrix multiplication needs to be written in the memory during the computation. Suppose that $A_1 X$ is computed first, and then $(A_1 X) A_2^\top$ is computed. By Lemma F.1, computing $(A_1 X)$ has I/O compleixty

$$\Omega(\frac{(\mathrm{nnz}(A_1) + \mathrm{nnz}(X)) \sqrt{\mathrm{nnz}(A_1 X)}}{\sqrt{M}}) \geq \Omega(\frac{2 \sqrt{\mathrm{nnz}(A_1) \mathrm{nnz}(X)} \sqrt{\mathrm{nnz}(A_1 X)}}{\sqrt{M}})$$

$$\geq \Omega(\frac{2 \sqrt{Z_A Z_X Z_{AX}}}{\sqrt{M}})$$

where the first step uses Cauchy-Schwarz inequality, the second step uses the definition of $Z_A, Z_X$ and $Z_{AXA}$.

By Lemma F.1, computing $(A_1 X) A_2^\top$ has I/O compleixty

$$\Omega(\frac{(\mathrm{nnz}(A_1 X) + \mathrm{nnz}(A_2)) \sqrt{\mathrm{nnz}(A_1 X A_2^\top)}}{\sqrt{M}}) \geq \Omega(\frac{\mathrm{nnz}(A_2) \sqrt{\mathrm{nnz}(A_1 X A_2^\top)}}{\sqrt{M}})$$

$$\geq \Omega(\frac{Z_A\sqrt{Z_{AXA}}}{\sqrt{M}}).$$

where the first step follows by the basic algebra, the second step uses the definition of $Z_A$ and $Z_{AXA}$. Therefore, computing $A_1XA_2^\top$ in this way has I/O complexity $\Omega(\frac{Z_A\sqrt{Z_{AXA}}+\sqrt{Z_AZ_XZ_{AX}}}{\sqrt{M}})$. Similary, suppose that $XA_2^\top$ is computed first. Then we can also get the I/O complexity $\Omega(\frac{Z_A\sqrt{Z_{AXA}}+\sqrt{Z_AZ_XZ_{AX}}}{\sqrt{M}})$.

Therefore, the sparse semi-ring matrix multiplication that computes $A_1XA_2^\top$ has I/O complexity $\Omega(\min\{\frac{Z_A^2+Z_AZ_X}{\sqrt{M}}, \frac{Z_A\sqrt{Z_{AXA}}+\sqrt{Z_AZ_XZ_{AX}}}{\sqrt{\sqrt{M}}}\})$. $\qquad\square$

Next, we can apply Lemma F.2 to get the lower bound of sparse attention forward and backward passes.

**Theorem F.3** (Lower bound for sparse attention forward). *Suppose $n$ is the input length, $d$ is the head dimension, and $M$ is the cache size. Let $Z_A := \min\{\mathrm{nnz}(A_1), \mathrm{nnz}(A_2)\}, Z_X := \mathrm{nnz}(X), Z_{AX} = \min\{\mathrm{nnz}(A_1X), \mathrm{nnz}(XA_2^\top)\}, Z_{AXA} := \mathrm{nnz}(A_1XA_2^\top)$. Then any algorithm for attention forward computation using sparse semi-ring matrix multiplication has I/O complexity $\Omega(\min\{\frac{Z_A^2+Z_AZ_X}{M}, \frac{Z_A\sqrt{Z_{AXA}}+\sqrt{Z_AZ_XZ_{AX}}}{\sqrt{M}}\})$.*

*Proof.* Any algorithm for attention forward computation needs to compute the matrix product $A_1XA_2^\top$ to obtain the attention matrix. Thus by applying Lemma F.2, we complete the proof. $\qquad\square$

**Theorem F.4** (Lower bound for sparse attention backward). *Suppose $n$ is the input length, $d$ is the head dimension, and $M$ is the cache size. Let $Z_A := \min\{\mathrm{nnz}(A_1), \mathrm{nnz}(A_2)\}, Z_X := \mathrm{nnz}(X), Z_{AX} = \min\{\mathrm{nnz}(A_1X), \mathrm{nnz}(XA_2^\top)\}, Z_{AXA} := \mathrm{nnz}(A_1XA_2^\top)$. Then any algorithm for attention backward computation using sparse semi-ring matrix multiplication has I/O complexity $\Omega(\min\{\frac{Z_A^2+Z_AZ_X}{M}, \frac{Z_A\sqrt{Z_{AXA}}+\sqrt{Z_AZ_XZ_{AX}}}{\sqrt{M}}\})$.*

*Proof.* Any algorithm for attention backward computation needs to compute the matrix product $A_1XA_2^\top$ to obtain the attention matrix. Thus by applying Lemma F.2, we complete the proof. $\qquad\square$

# G BROADER IMPACTS

This paper presents work whose goal is to understand the theory of attention mechanisms. Our findings provide a theoretical foundation for designing efficient algorithms that improve the scalability and performance of modern AI systems. Although the primary contributions are technical, this work has the potential to impact a broad range of applications, from accelerating model training and inference to enabling resource-efficient deployment in real-world settings. While our work has many potential societal consequences, including advancements in natural language understanding and accessibility of AI technologies, none of which we feel must be specifically highlighted here.

## LLM USAGE DISCLOSURE

LLMs were used only to polish language, such as grammar and wording. These models did not contribute to idea creation or writing, and the authors take full responsibility for this paper's content.

