# OpenReview forum: "On Fine-Grained I/O Complexity of Attention Backward Passes"
_ICLR.cc/2026/Conference — Submitted to ICLR 2026_

### Official Review · Reviewer_KW4D · 2025-10-30

**Soundness:** 4
**Presentation:** 3
**Contribution:** 1
**Rating:** 2
**Confidence:** 5

**Summary:**

The authors consider the I/O complexity of Attention gradient computation. In hardware, data is typically arranged hierarchically, with data stored in an unbounded memory, and computation occurring in a bounded cache. To compute, data is moved into the cache, computation occurs, and the result is saved in memory. Since data movement is typically more expensive that computation, I/O complexity measures only the data movements. The goal of I/O complexity is to design algorithms minimizing I/Os. Given the prevalence of attention and the success of the FlashAttention algorithm, it is a practically important question to understand whether the training process can be optimized w.r.t. I/O complexity.

The authors give I/O optimal bounds for the computation of attention gradient when restricted to algorithms using standard matrix multiplication. The authors also consider sparse attention, and give lower bounds for algorithms using standard matrix multiplication in this setting.

While the statement of the main result is interesting, the techniques are identical to prior work, and in fact the main result can be obtained immediately from the lower bound for the forward pass. Furthermore, the lower bound on sparse attention is not well substantiated without a matching upper bound (or at least some improvement over the naive algorithm). Thus I recommend reject.

**Strengths:**

The authors study a practically interesting problem, and give tight results.

They initiate the study of sparse I/O attention.

**Weaknesses:**

The main result (lower bound for attention gradient computation) is essentially immediate from prior work. In particular, a previous paper proves that any algorithm that computes the attention matrix already requires the FlashAttention lower bound. Since attention gradient computation involves a n x d and d x n matrix product, this immediately implies the desired lower bound. Similarly, the new upper bound for gradient computation in the small cache setting is a consequence of the equivalence with matrix multiplication (the easy direction - using matrix multiplication we can compute attention gradients).

The sparse attention lower bound is not well motivated if there is no matching upper bound, or at least some improvement on the trivial algorithm. Even if this is hard to prove, there should be some discussion towards what the obstacles are.

**Questions:**

What are the main obstacles towards designing I/O efficient algorithms for sparse attention?

---

> ### Author Response · Authors · 2025-11-18
> **First response for Reviewer KW4D**
>
> Thanks for your constructive and in-depth feedback! We hope our clarifications help address the concerns you raised.
>
> ### **Weakness 1: Difference compared to [Saha and Ye, 2024]**
>
> The backward computation of attention is highly non-trivial and involves multiple interconnected steps. Our contribution is to explicitly derive and abstract the full backward computational graph, identify all intermediate dependencies, and rigorously show where the true I/O bottleneck lies. In particular, it is true that we prove that the critical bottleneck is indeed the matrix multiplication component, but this conclusion is not immediate, since it requires constructing the complete multi-phase backward computation graph and applying careful analysis to each stage. This detailed characterization, along with the tight lower bounds and matching upper bounds in different cache regimes, constitutes the core technical contribution of our work.
>
> To clarify, we also summarize the novelty of our work here:
> - **Small cache regime**: Our algorithm for the small-cache setting is not a trivial modification of existing approaches. It requires careful orchestration of four distinct phases with specific block decompositions that differ fundamentally from FlashAttention’s strategy. Our Theorem 4.4 provides the first tight lower bound for $M = o(d^2)$, requiring novel reduction arguments to matrix multiplication that do not appear in [Saha and Ye, 2024].
> - **Sparse attention bounds**: We extend the pebble-game framework to sparse attention (Theorem 4.5), providing fine-grained upper and lower bounds for sparse attention across all cache sizes, which is not addressed in [Saha and Ye, 2024].
> - **Regarding $A_1^\top p(X) A_2$ complexity**: While this operation does involve matrix multiplications, the challenge lies in computing $p(X)$ (which depends on the Softmax gradients) efficiently in the small-cache regime.
>
>
> ### **Weakness 2 and Question 1: Novelty of sparse attention results**
>
> Thank you for the insightful comment. We would like to clarify the novelty of our sparse-attention results. Our contributions are not straightforward extensions and provide new theoretical insights.
>
> 1. **FlashAttention bounds may not be optimal in sparse settings.**
>    The runtime of the sparse implementation of FlashAttention is $O(Nd + \frac{N^2 d^2}{M} s)$.
>    Even though the block-sparsity factor $s$ may be small in practice, this bound still fundamentally depends on the dense-size term $Nd$, regardless of the actual sparsity pattern. In contrast, our lower bound depends directly on the input sparsity, characterized by the number of nonzero entries in the input and in the query-key matrices. Our analysis therefore reveals that FlashAttention may not be optimal in the sparse setting and highlights the need for truly sparsity-aware algorithmic designs.
>
> 2. **Our sparse results form a unified and more general framework.**
>    Notably, our sparse bounds subsume the dense-attention bounds as a special case: when the matrices become dense, our bounds recover the dense setting results (Remark 4.6), demonstrating internal consistency and generality. We also generalize the cache threshold to the sparse setting (Remark 4.7), providing a fine-grained characterization of how sparsity interacts with different cache regimes.

---

> > ### Comment · Reviewer_KW4D · 2025-11-25
> >
> > Thank you for the rebuttal. I remain negative overall about the paper and will explain why below.
> >
> > Lower Bound Novelty: while constructing the computational graph involves effort, it is not necessary. It is clear any attention computation must compute A_1 X A_2 which already invokes the lower bound (in fact you prove Lemma F.3 this way).
> >
> > Upper Bound: similarly, every computation is either a matrix product of dimension n x d x n or an operation eg soft max that is linear in IO. Thus, it is clear that efficient matrix multiplication implies efficient Attention (as is observed in previous work).
> >
> > The one thing that is novel in my view is the result on Sparse Attention. I think this is a good problem to consider but the single lower bound is not enough to warrant acceptance (and in fact the authors treat it as a side result). I would like to see stronger lower bounds without assumptions on the matrix multiplication algorithm, and most importantly more efficient algorithms than FlashAttention in the sparse setting.

---

### Official Review · Reviewer_3EBW · 2025-10-31

**Soundness:** 4
**Presentation:** 3
**Contribution:** 2
**Rating:** 4
**Confidence:** 2

**Summary:**

The paper extends the analysis of I/O complexity of exact attention appearing in [Dao, 2022] and [Saha & Ye, 2024], specifically providing tight bounds on the I/O complexity of the attention backwards pass using the red-blue pebble game framework. The results suggest that the popular FlashAttention algorithm is optimal in both forwards and backwards modes in the large cache regime (most practically relevant), while providing an improved algorithm in the small cache regime. The authors also extend the analysis to the sparse attention regime.

**Strengths:**

- The paper's derivations seem to be solid and rigorous, to the best of my understanding.
- The paper extends the results appearing in the previous work, thus completing the I/O complexity analysis for both forwards and backwards passes, small and large cache regimes, as well as dense and sparse attention.
- The paper is well-written and easy to follow.

**Weaknesses:**

Overall, the paper seems to be a direct extension of [Saha & Ye, 2024], adding tight bounds for the I/O complexity of attention backwards pass. However, the results seem to directly mirror the prior work; the authors utilise the same framework, and provide similar asymptotic bounds and conclusions. Due to this, my impression is that the work, although mathematically solid, seems to be incremental. The small-cache algorithm, as well as theoretical derivations seem to follow directly from [Saha & Ye, 2024], and from the practical perspective do not offer a significant contribution (as noted in the paper, the large- cache regime is more practically relevant, and FlashAttention is proven to be optimal). Due to this, my impression is that the scope of the paper is not quite sufficient for publication in ICLR.

**Questions:**

- Could the authors clarify how their small-cache algorithm differs/complements the similar proposition from [Saha & Ye, 2024]?

---

> ### Author Response · Authors · 2025-11-18
> **First response for Reviewer 3EBW**
>
> Thank you for your thoughtful comments. We hope our clarifications help address the concerns you raised.
>
> ### **Weakness 1 and Question 1: Difference compared to [Saha and Ye, 2024]**
>
> **1. Novelty**:
>
> First, we would like to clarify our novelty compared to [Saha and Ye, 2024]. Our contribution in the backward pass is non-trivial and technically novel. It is true that we similarly use the red-blue pebble game as a foundation, as it is the standard model for analyzing I/O complexity. However, our contribution lies in new constructions and problem-specific adaptations:
>
> - **Computational graph complexity**: The backward pass requires computing gradients through the entire attention mechanism, involving multiple interdependent matrix operations (computing $p(X)$, handling Softmax derivatives, etc.) that do not exist in the forward pass. We provide the exact computational graph (Figure 2), introducing multi-phase backward computation graphs with hierarchical dependencies not previously modeled in [Saha and Ye, 2024], which is a necessary condition for complexity analysis.
> - **Small cache regime**: Our algorithm for the small-cache setting is not a trivial modification of existing approaches. It requires careful orchestration of four distinct phases with specific block decompositions that differ fundamentally from FlashAttention’s strategy. Our Theorem 4.4 provides the first tight lower bound for $M = o(d^2)$, requiring novel reduction arguments to matrix multiplication that do not appear in [Saha and Ye, 2024].
> - **Sparse attention bounds**: We extend the pebble-game framework to sparse attention (Theorem 4.5), providing fine-grained upper and lower bounds for sparse attention across all cache sizes, which is not addressed in [Saha and Ye, 2024].
> - **Regarding $A_1^\top p(X) A_2$ complexity**: While this operation does involve matrix multiplications, the challenge lies in computing $p(X)$ (which depends on the Softmax gradients) efficiently in the small-cache regime.
>
> Thus, while [Saha and Ye, 2024] inspired our framework, our results go beyond by addressing new theoretical challenges unique to backward computation. Therefore, while the tool is the same, the application is original and tailored to new theoretical questions.
>
> **2. Importance of the small cache setting**
>
> Second, we would like to emphasize that the small-cache regime is valuable and worth discussing, highlighting this paper’s practical relevance. The small-cache analysis remains highly important because, as models continue to grow larger and are increasingly deployed on edge devices, the effective cache size may often be small relative to the hidden dimension. In such settings, our small-cache algorithms become directly relevant. Thus, our work provides theoretical insights that can guide future developments in attention mechanisms under evolving model sizes and hardware constraints.

---

> > ### Comment · Reviewer_3EBW · 2025-11-26
> >
> > I'd like to thank the authors for their response to my comments and for highlighting the contributions of their paper compared to [Saha and Ye, 2024]. I appreciate the additional context regarding the backward-pass graph structure, the technical aspects of the small-cache regime, and the extensions to sparse attention.
> >
> > That said, after considering your clarifications, I am still inclined to remain with my original assessment. While I agree that the backwards pass construction, as well as that the small-cache extension are important and take care (I indeed highlighted these as strengths in my review), I am still concerned that the incremental nature of these contributions, in the absence of empirical demonstrations, limits the overall impact of the paper. The sparse attention results indeed could be of interest to the community, but again without a practical demonstration of a speed-up against FlashAttention, it is difficult to fully assess their importance.

---

### Official Review · Reviewer_qFFE · 2025-10-31

**Soundness:** 4
**Presentation:** 3
**Contribution:** 4
**Rating:** 8
**Confidence:** 3

**Summary:**

The paper analyzes the I/O (cache ↔ memory) complexity of the backward pass of exact softmax attention under standard GEMM, using the red–blue pebble framework. It proves *matching upper and lower bounds across all cache sizes*, with a phase transition at ($M = \Theta(d^2)$) ($M$ is the cache size and $d$ is attention head dimension).

In the large-cache regime ($M=\Omega(d^2)$), the bounds match FlashAttention’s behavior and establish optimality; in the small-cache regime ($M=o(d^2)$), the paper gives a strictly better algorithm (and matching lower bound) than FlashAttention. It also gives lower bounds for sparse attention, recovering the dense case as a special case.

**Strengths:**

### Originality

* Provides the first matching upper and lower bounds for the backward pass of exact attention for all cache sizes with a clean phase transition at ($M=\Theta(d^2)$) (Theorem 1.1).
* Extends to sparse attention with lower bounds that recover the dense case as a special instance.

### Quality

* Uses the red–blue pebble framework rigorously and states Theorem 1.1 with an explicit formula covering both regimes.
* Gives matching bounds in each regime: large-cache upper (Thm 4.1) and lower (Thm 4.2), small-cache upper via Algorithm 6 (Thm 4.3) and lower (Thm 4.4).

### Clarity.

* Figure 1 clearly contrasts the paper’s tight bound (red) with FlashAttention’s upper bound (blue dashed) and marks the cross-point ($M=\Theta(d^2)$).
* Theorems in §4 are presented as informal versions which helped readabillity.

### Significance

* In the large-cache regime, results match FlashAttention and establish optimality; in the small-cache regime, Algorithm 6 is provably better than FlashAttention.

**Weaknesses:**

1. **Positioning vs prior work could be tighter.** The paper clearly cites Dao et al. (FlashAttention) and Saha & Ye for forward-pass tightness; it mentions Addanki et al. (streaming/approximate attention) in related work, but a compact comparison table clarifying different problem settings (exact vs approximate, streaming vs two-level memory) would help readers situate novelty.

2. **Practical relevance narrative.** The paper *does* discuss when small-cache arises (e.g., per-SM caches on older GPUs) and even gives A100 vs GTX1060 examples; expanding this with a short table of device-level (M) estimates and typical head sizes (d) would strengthen the “why it matters” section.

**Questions:**

1. **Scope vs Addanki et al. (2023).** Please add a small table clarifying the differences (objective: exact vs approximate; model: two-level I/O vs streaming; bounds reported) and why your results are not directly comparable numerically.

2. **Multi-head attention.** Your bounds are given per head; what changes (if any) under (H) heads computed in parallel. Does tiling across heads alter the asymptotics or only the constants?

3. **Device checklist.** Consider adding a table (SM/L1 size, datatype, typical ($d$)) for a few GPUs/edge devices to show where ($M \lessgtr d^2$) actually falls.

---

> ### Author Response · Authors · 2025-11-18
> **First response for Reviewer qFFE**
>
> Thanks for your strong recommendation of our work! We would like to take this opportunity to address your questions and clarify several points.
>
> ### **Weakness 1 and Question 1: Comparison with FlashAttention, [Saha and Ye, 2024], and [Addanki et al., 2023]**
>
> **Comparison to FlashAttention and [Saha and Ye, 2024]**: The comparison of our work and these two works can be found in Table 1 on page 3 of our work, and we would also like to restate our novelty here:
>
> - **Computational graph complexity**: The backward pass requires computing gradients through the entire attention mechanism, involving multiple interdependent matrix operations (computing $p(X)$, handling Softmax derivatives, etc.) that do not exist in the forward pass. We provide the exact computational graph (Figure 2), introducing multi-phase backward computation graphs with hierarchical dependencies not previously modeled in [Saha and Ye, 2024], which is a necessary condition for complexity analysis.
> - **Small cache regime**: Our algorithm for the small-cache setting is not a trivial modification of existing approaches. It requires careful orchestration of four distinct phases with specific block decompositions that differ fundamentally from FlashAttention’s strategy. Our Theorem 4.4 provides the first tight lower bound for $M = o(d^2)$, requiring novel reduction arguments to matrix multiplication that do not appear in [Saha and Ye, 2024].
> - **Sparse attention bounds**: We extend the pebble-game framework to sparse attention (Theorem 4.5), providing fine-grained upper and lower bounds for sparse attention across all cache sizes, which is not addressed in [Saha and Ye, 2024].
> - **Regarding $A_1^\top p(X) A_2$ complexity**: While this operation does involve matrix multiplications, the challenge lies in computing $p(X)$ (which depends on the Softmax gradients) efficiently in the small-cache regime.
>
> **Comparison to [Addanki et al., 2023]**: Our setting and goals differ fundamentally from those of streaming attention, so the two works are not directly comparable. I/O complexity is studied under a two-level memory-hierarchy model where the system has sufficient total memory but experiences a performance gap between fast memory (e.g., SRAM) and slow memory (e.g., HBM). The objective is to minimize data movement between these two levels, which is the principal bottleneck in modern GPUs.
>
> In contrast, streaming attention assumes an extremely memory-constrained setting (e.g., $O(\sqrt{n})$ total memory) and focuses on designing approximate algorithms that operate under strict space limits. Their work does not analyze data-movement costs or the I/O behavior of the memory hierarchy.
> Beyond the modeling differences, the technical scope also differs:
> - The streaming-attention algorithms in [Addanki et al., 2023] are approximate, whereas our algorithms and bounds are exact.
> - Their work analyzes only the forward pass, while our results cover both the forward and backward passes with tight upper and lower bounds.
> For these reasons, our contributions address a distinct theoretical question and operate under a fundamentally different computational model.
>
> ### **Weakness 2 and Question 3: Practicality of the small-cache setting**
>
> The current network architectures usually set $d = 128$. In this case, the dividing point is approximately $d^2 \times$ size_of(data type), e.g., $16,384 \times 32$-bit = $65.5$ KB for float32. For the NVIDIA A100 GPU, the size of each streaming multiprocessor (SM/L1 cache) is $192$ KB, so we should choose FlashAttention. However, for old GPUs such as NVIDIA GTX1060, the size of each SM is $48$ KB, so the algorithm for the small cache size is suitable.
> Furthermore, the small-cache analysis remains highly valuable because, as models continue to grow larger and are increasingly deployed on edge devices, the effective cache size may often be small relative to the hidden dimension. In such settings, our small-cache algorithms become directly relevant. Thus, our work provides theoretical insights that can guide future developments in attention mechanisms under evolving model sizes and hardware constraints.
>
> ### **Question 2: Multi-head attention**
>
> Our bounds naturally extend to multi-head attention. Each attention head performs an independent attention computation with its own query, key, and value matrices, and there is no data dependency across heads. Therefore, the I/O cost for multi-head attention is simply the sum of the costs of the individual heads. If sufficient GPU cores and fast memory (i.e., cache) are available, these heads can be computed fully in parallel, yielding the same per-head I/O complexity as in the single-head case.
>
> Thank you again for your insightful review and for recommending acceptance. We hope that the above clarifications address your concerns. If you believe these responses resolve the issues you raised, we would be happy to incorporate them into the next version of our PDF.

---

### Official Review · Reviewer_gi5z · 2025-11-04

**Soundness:** 3
**Presentation:** 3
**Contribution:** 1
**Rating:** 6
**Confidence:** 3

**Summary:**

The original FlashAttention paper provides upper I/O complexity bounds for the backward pass of the exact attention computation, but does not provide lower bounds. This raises the question: what is the optimal I/O complexity of the attention backward pass? This paper provides a lower bound as a function of cache size. Interestingly, they show that the lower bound changes at a crossover point where the cache size if $o(d^2)$.

**Strengths:**

- The authors show that there is room at small cache sizes, to potentially provide a speedup over FlashAttention by reducing I/O complexity.
- The paper is pretty easy to follow and does quite a good job situating itself with respect to prior work.

**Weaknesses:**

- The authors do not provide an implementation of their algorithm, and so they cannot demonstrate that it actually provides a speedup over FlashAttention. The claim that the “algorithm designed for small cache sizes would become relevant and useful”, is speculative. In my view, this is the most significant limitation of this work.
- The result is only applicable for very small cache sizes, and does not apply to modern GPUs typically used for training (A100s, H100s, B200s).
- This paper (like prior work before it) assume a two-level memory hierarchy. This may limit the applicability of the results, especially since newer chips include more complex memory hierarchies including

**Questions:**

- Does Algorithm 6 increase the FLOPs required — even if only by a constant factor?
- Can the authors provide an implementation of their algorithm and demonstrate that it can provide a speed up on GPUs with small cache sizes?

---

> ### Author Response · Authors · 2025-11-18
> **First response for Reviewer gi5z, part 1/2**
>
> We appreciate the positive evaluation of our work and would like to take this opportunity to clarify a few points.
>
> ### **Weakness 1 & Question 2: Lack of experiments**
> We agree that additional experiments could make the paper more comprehensive. However, implementing our approach may pose several substantial challenges, including:
> - Current CUDA implementations combine various optimizations beyond pure I/O considerations.
> - GPU hardware abstracts away many low-level memory operations, making it difficult to precisely measure theoretical I/O complexity.
> - The theoretical model considers an idealized two-level memory hierarchy, while real GPU memory hierarchies are more complex.
>
> Furthermore, we would like to clarify that this work is purely theoretical, focusing on establishing fundamental bounds on I/O complexity for attention mechanisms. I/O complexity is a well-established topic in complexity theory and TCS [1,2], and recent ML conferences have also accepted many purely theoretical works on LLM and attention complexity [3,4,5], where a notable I/O complexity theory paper is [3]. The key contributions of this paper include:
> - Providing rigorous mathematical proofs for tight bounds on I/O complexity.
> - Identifying the critical point where I/O complexity behavior changes.
> - Establishing theoretically optimal algorithms in different cache regimes.
> - Extending the analysis to sparse attention with fine-grained bounds.
>
> Our theoretical foundations can guide future practical implementations and remain valid and valuable regardless of hardware, with bridging theory and practice noted as future work.
> ### **Weakness 2: Limited to small cache sizes**
> Thanks for your comments. The small-cache analysis remains highly valuable because, as models continue to grow larger and are increasingly deployed on edge devices, the effective cache size may often be small relative to the hidden dimension. In such settings, our small-cache algorithms become directly relevant. Thus, our work provides theoretical insights that can guide future developments in attention mechanisms under evolving model sizes and hardware constraints.
>
> Furthermore, we would also like to clarify that our work is not limited to small cache sizes. This is a comprehensive theoretical study of the I/O complexity of attention computation. In addition to the small-cache results in Theorems 4.3-4.4, we also provide large-cache results in Theorems 4.2 and 4.5, including the general-attention backward lower bound and the sparse-attention upper and lower bounds, which are novel contributions of this paper.
> ### **Weakness 3: Two-level memory hierarchy**
> Thanks for pointing this out. The two-level memory hierarchy is a standard and widely used abstraction in both theoretical works [1,2,3] and well-known empirical studies such as the FlashAttention series [6,7,8]. As discussed in these papers, the primary performance bottleneck in modern GPUs is the data transfer cost between SRAM and HBM, making the two-level model an appropriate and commonly adopted representation of GPU memory behavior.
>
> A concrete example can be seen in the NVIDIA H100 GPU (https://developer.nvidia.com/blog/nvidia-hopper-architecture-in-depth/), where the fast on-chip memory is the 50 MB L2 cache (corresponding to SRAM in the FlashAttention papers), and the slower but much larger memory is the 80 GB HBM system. This directly aligns with the two-level hierarchy assumed in our theoretical model.
>
> ### **Question 1: Clarification on Algorithm 6**
> In I/O complexity studies, including our work, the FLOPs do not increase, since the underlying computations remain exactly the same. Our algorithm only changes the scheduling between computation and data movement to reduce memory exchanges across different memory spaces (e.g., fast cache such as SRAM and slow memory such as HBM). The improvement comes from reduced I/O time, which is the practical performance bottleneck, as demonstrated in a series of works on this topic [6,7,8].
>
> We sincerely thank you once more for your encouraging recommendation and hope that our response has adequately addressed your concerns.

---

> ### Author Response · Authors · 2025-11-18
> **First response for Reviewer gi5z, part 2/2**
>
> ### **References**
> [1] Hong Jia-Wei, H. T. Kung. “I/O complexity: The red-blue pebble game”. STOC 1981.
>
> [2] Grey Ballard, James Demmel, Olga Holtz, Oded Schwartz. “Graph Expansion and Communication Costs of Fast Matrix Multiplication”. JACM 2013.
>
> [3] Barna Saha, Christopher Ye. “The I/O Complexity of Attention, or How Optimal is Flash Attention?”. ICML 2024.
>
> [4] Angeliki Giannou, Shashank Rajput, Jy-Yong Sohn, Kangwook Lee, Jason D. Lee, Dimitris Papailiopoulos. “Looped transformers as programmable computers”. ICML 2023.
>
> [5] Josh Alman, Zhao Song. “The fine-grained complexity of gradient computation for training large language models”. NeurIPS 2024.
>
> [6] Tri Dao, Dan Fu, Stefano Ermon, Atri Rudra, Christopher Ré. “FlashAttention: Fast and Memory-Efficient Exact Attention with IO-Awareness”. NeurIPS 2022.
>
> [7] Tri Dao. “FlashAttention-2: Faster Attention with Better Parallelism and Work Partitioning”. ICLR 2024.
>
> [8] Jay Shah, Ganesh Bikshandi, Ying Zhang, Vijay Thakkar, Pradeep Ramani, Tri Dao. “FlashAttention-3: Fast and Accurate Attention with Asynchrony and Low-precision”. NeurIPS 2024.

---

### Author Response · Authors · 2025-12-03
**Summary of the discussion period**

Dear Area Chair,

Given the unprecedented situation regarding the OpenReview information leak and ICLR’s decision to revert pre-rebuttal scores and reassign ACs, we would like to summarize our interactions with the reviewers during the discussion period to ensure that our responses are clearly documented.

- **Reviewer gi5z**: This reviewer asked about (1) the absence of experiments in a theory paper, (2) the relevance of the small-cache setting, (3) practicality of the two-level memory hierarchy, and (4) clarification on Algorithm 6. We addressed each point in detail, explaining that purely theoretical papers without experiments are common in ML theory, and that our setting matches real LLM computation. The reviewer **maintained a positive score of 6**.

- **Reviewer qFFE**: This reviewer asked about (1) novelty relative to FlashAttention, Saha & Ye (2024), and Addanki et al. (2023), (2) the importance of the small-cache regime, and (3) generalization to multi-head attention. We clearly compared our work to prior literature and highlighted the universality and practical relevance of our theory. The reviewer **maintained a positive score of 8**.

- **Reviewer 3EBW**: This reviewer raised concerns about (1) the lack of experiments and (2) novelty compared with Saha & Ye. We addressed both questions carefully in the rebuttal, and the reviewer **maintained their score**, noting that our sparse-attention results are interesting.

- **Reviewer KW4D**: This reviewer asked about novelty relative to Saha & Ye and our contributions to sparse attention. We provided a clear comparison, and the reviewer **maintained their score**, acknowledging that our sparse-attention result is novel.

We believe we have thoroughly addressed all reviewer comments during the discussion period. We hope this summary clarifies the reviewers’ positions following the rebuttal and highlights our efforts to engage constructively with their feedback.

The primary and recurring concern in the reviews is the absence of experiments. We would like to respectfully note that many recent ML conferences have accepted purely theoretical works on LLM and attention complexity. Our paper studies an important problem, introduces non-trivial new theoretical results on attention backward computation and sparse attention, and holds strong potential for guiding real LLM inference. Given the divergence in reviews, we kindly ask for your careful and thoughtful consideration of our submission.

Thank you again for your time and consideration.

---

### Meta-Review · Area_Chair_7UuA · 2025-12-21

**Summary:**

This submission studies the I/O complexity of exact softmax-attention backward (gradient) computation under the red–blue pebble framework, with additional results for sparse attention. Reviewers generally agree the proofs are technically solid and the paper is clearly written, but they disagree on whether the contribution rises to the bar for ICLR. The discussion converged on two recurring decision-driving concerns: (i) limited practical substantiation (no implementation/experiments) despite the paper’s systems-motivated framing, and (ii) perceived incremental novelty, especially relative to Saha & Ye (2024) and the known link between attention computation and matrix multiplication I/O bounds. While some reviewers view the tight bounds across cache regimes as a meaningful theoretical completion, others—particularly the most confident negative reviewer—argue the main lower/upper bounds follow almost immediately from prior results and that the new material is insufficiently developed beyond a sparse-attention lower bound.

**Reviewer Concerns:**

**Concerns addressed by the rebuttal (partially / sufficiently for some reviewers):**

(1) The authors clarified differences in scope (backward vs forward; exact vs approximate; two-level I/O vs streaming) and pointed to missing/desired comparative framing. This seemed to satisfy the more positive reviewers, though not the negative ones.

(2) The authors defended the abstraction as standard for I/O complexity and consistent with the SRAM/HBM bottleneck framing used in FlashAttention-style analyses. This addressed the concern at the modeling level, though it does not resolve the separate question of practical impact.

(3) The authors argued the per-head bounds extend by additivity and parallelism does not change asymptotics—reasonable and largely uncontested.

(4) The authors provided examples (older GPUs / edge settings) and argued future relevance as deployment diversifies. This explanation was accepted by supportive reviewers but remained secondary to the novelty/impact debate for skeptical reviewers.




**Concerns still outstanding after the rebuttal (key reasons for rejection):**

***(1) Novelty/Incrementality of the main result:***

Reviewer KW4D (high confidence) maintains that the backward-pass lower bound and much of the story is essentially immediate from prior work (attention requires an $n \times d$ by $d \times n$ multiplication / computing attention-like products already triggers the known lower bound), and that constructing the full backward computational graph is not necessary to obtain the stated result.

Reviewer 3EBW also remains concerned the paper largely mirrors the prior framework and conclusions, making the contribution feel incremental for ICLR even if technically correct.

The rebuttal clarified what the authors view as non-trivial (graph construction, phase orchestration in small-cache, reductions), but it did not convincingly change the skeptics’ view that the result follows directly from known components.


***(2) Practical impact / absence of empirical validation:*** While this is a theory paper, multiple reviewers still found the lack of any implementation evidence limiting given the paper’s systems-motivated framing (“relevance/usefulness” and comparison to FlashAttention). The rebuttal argues experiments are hard and not required for theory; this is true in general, but it does not resolve the reviewers’ concern that the claimed practical relevance—especially for sparse attention and small-cache regimes—remains speculative in the current submission.


***(3) Sparse attention results are underdeveloped:*** Even the negative reviewer acknowledges the sparse-attention direction is the most novel aspect, but strongly argues that a lone lower bound without a matching upper bound or a demonstrably better algorithm than naive/FlashAttention-style baselines is not enough. The rebuttal did not provide a concrete algorithmic advance for sparse attention nor a clear roadmap of obstacles that would justify the current partial result as publication-ready.

**Reviewer Scores:**

gi5z: Likely stays at 6 (marginal accept; okay with reject). The rebuttal addressed modeling questions and framed the work as theory, but the implementation gap remains.

qFFE: Likely stays at 8 (accept/poster). Concerns were mainly positioning and presentation; rebuttal addressed them well.

3EBW: Likely stays at 4 (marginal reject). The reviewer explicitly reiterated that, despite appreciating clarifications, the work remains too incremental without empirical demonstration.

KW4D: Likely stays at 2 (reject). The reviewer explicitly remains negative, with high confidence, arguing the main results follow immediately from prior work and that sparse attention is not developed enough to carry the paper.

---

### Decision · Program_Chairs · 2026-01-26

Reject